# Improving LLM Symbolic Problem-Solving via Automated Heuristic Discovery

## Abstract

We consider enhancing large language models (LLMs) for symbolic problem-solving tasks. While existing inference-time techniques let LLMs explore intermediate steps for problem-solving, they rely on noisy self-verification or external verifiers, which demand significant data and computations. Here, we propose Automated Heuristics Discovery (AutoHD), a novel approach that enables LLMs to generate heuristic functions to guide inference-time search through accurate evaluation of intermediate steps. These heuristic functions are further refined through an evolution process, improving their robustness and effectiveness. Our method requires no additional model training or fine-tuning, and the explicit definition of heuristic functions provides interpretability and insights into the solving process. Extensive experiments on diverse tasks demonstrate significant gains over multiple baselines, including nearly twice the accuracy on some tasks, establishing AutoHD as a reliable and interpretable solution.

## 1 Introduction

Large language models (LLMs) are increasingly being applied to tasks that require structured reasoning and decision-making, extending beyond traditional NLP applications such as translation and summarization. These models have demonstrated potential in complex reasoning tasks, including arithmetic problem-solving (Cobbe et al., 2021), logical reasoning (Saparov & He, 2023), vision-based reasoning (Parashar et al., 2024), and multi-step problem-solving (Yang et al., 2018). Building on these strengths, researchers have begun using LLMs for symbolic problem-solving tasks, which require sequential decision-making, exploration of intermediate steps, and strategic thinking. To further enhance LLMs' capabilities in such tasks, test time inference techniques (Wei et al., 2022; Wang et al., 2022; Hao et al., 2023) like Tree-of-Thought (Yao et al., 2024) enable LLMs to search over the intermediate steps, improving their ability to arrive at correct solutions. From solving puzzles to generating action plans and strategies, showcasing their potential as reliable tools for tackling challenges that go beyond traditional NLP applications.

While test-time inference techniques have achieved notable success, there remain opportunities to improve them. *For instance*, early *techniques*, such as Chain-of-Thought (Wei et al., 2022), rely on a linear reasoning process, which often fails to explore diverse solution paths or recover from errors introduced during intermediate steps. This limitation may result in suboptimal performance, particularly in symbolic problem-solving tasks (Valmeekam et al., 2022) where early mistakes tend to propagate and result in incorrect final outcomes. More recent methods have attempted to address these challenges by incorporating mechanisms to search over intermediate steps and verify their correctness, using either LLM self-verification (Hao et al., 2023) or external models to evaluate these steps (Kambhampati et al., 2024). However, self-verification has been shown to be unreliable, as LLMs cannot reliably identify errors (Huang et al., 2023). On the other hand, approaches that require additional model training come with significant costs in terms of data and computational resources *(Yu et al., 2024)*, making them less practical for many real-world applications.

Humans, on the other hand, rely on heuristics to simplify decision-making when faced with complex problems (Hjeij & Vilks, 2023; Simon, 1997; Tversky & Kahneman, 1974). *For example, when planning a route across multiple destinations, one might prioritize visiting nearby locations first, rather than exhaustively evaluating all possible permutations. Such heuristics help guide decision-making by enabling effective action prioritization.* Inspired by this human problem-solving behavior,

we propose automated heuristics discovery (AutoHD), an approach to help make correct decisions during problem-solving. Specifically, AutoHD prompts LLMs to explicitly generate reliable heuristic functions represented as Python code. During inference, intermediate steps *and their resulting states are evaluated using the generated heuristic function, guiding the search to the goal by estimating how far each state is from it.* To further enhance the robustness and performance of these heuristic functions, we introduce a heuristic evolution process that iteratively refines them over time. Notably, our approach achieves these advancements without requiring additional models or fine-tuning the pre-trained LLMs, making it lightweight and broadly applicable. Moreover, by explicitly defining the heuristic functions, AutoHD offers interpretability, allowing us to understand why LLMs prefer certain intermediate steps over others. Extensive experiments over three symbolic problem-solving tasks, including Blocksworld Valmeekam et al. (2022), Rubik's Cube (Ding et al., 2023), and the Game of 24 (Yao et al., 2024), demonstrate that AutoHD improves the performance of LLMs.

## 2 BACKGROUND AND RELATED WORK

We define a symbolic problem as $\{\mathcal{S}, s_0, \mathcal{G}, \mathcal{A}_\theta, T_\theta\}$. Here $\mathcal{S}$ represents the state space, containing all possible states. The initial state is denoted as $s_0 \in \mathcal{S}$, and the goal states are represented as $\mathcal{G} = \{g_0, g_1, \cdots, g_m\}$, where each $g_i \in \mathcal{S}$ is a possible goal state. Note, in some cases, there may be only one goal state, i.e., $\mathcal{G} = \{g_0\}$. The action space $\mathcal{A}_\theta(s)$ is generated by a pre-trained LLM with parameters $\theta$ based on the current state $s$, providing a set of valid actions for state $s$. The transition function $T_\theta : \mathcal{S} \times \mathcal{A} \to \mathcal{S}$, also parameterized by the LLM, predicts the outcome of applying an action $a \in \mathcal{A}_\theta(s)$ to a state $s$. A solution to the task is represented as a state-action trace $\Pi = (s_0, a_1, s_1, a_2, \cdots, a_n, s_n)$ where $s_{i+1} = T_\theta(s_i, a_{i+1})$ and $s_n \in \mathcal{G}$.

**LLMs for Problem Solving.** LLMs have demonstrated potential for reasoning and common-sense capabilities. Specifically, LLMs have been widely used as policy models for decision-making and problem-solving in diverse interactive environments, including robotics (Driess et al., 2023; Huang et al., 2022; Kannan et al., 2024; Singh et al., 2023), multimodal games (Fan et al., 2022; Wang et al., 2023), and text-based environments (Liu et al., 2023; Yao et al., 2022; Shinn et al., 2024). To further enhance their performance, researchers have developed test-time inference techniques that adaptively guide the model's reasoning and decision-making. For example, Chain of Thought (CoT) prompting (Wei et al., 2022) guides LLMs to generate intermediate reasoning steps to solve problems. Building on this, Tree of Thoughts (ToT) (Yao et al., 2024) and Graph of Thoughts (GoT) (Besta et al., 2024) explore multiple paths to improve decision-making and robustness. Similarly, XoT (Ding et al., 2023) trains an auxiliary policy model to assign rewards to solutions. Techniques like RAP (Hao et al., 2023) utilize Monte Carlo Tree Search to navigate the expansive action space, while FoR (Yu et al., 2024) finetunes the LLM to discover diverse and creative solutions across multiple tasks. Despite their effectiveness, these methods often rely on LLM self-evaluation, which can be noisy or incorrect (Huang et al., 2023; Stechly et al., 2024), or require additional fine-tuning, which has significant computational overhead.

**LLMs for Automated Heuristic Discovery.** Automatic heuristic design refers to the process of generating, optimizing, or adapting heuristic functions automatically to solve problems. Traditional heuristic functions are typically handcrafted based on domain expertise, but automatic methods use algorithms to create or refine these heuristic functions without manual intervention. Evolutionary algorithms like genetic programming (Koza, 1994) have been widely used, allowing heuristic functions to evolve through processes inspired by natural selection. Recently, LLMs have emerged as powerful tools for code generation. Their extensive pretraining provides them with broad knowledge and contextual understanding, enabling them to assist in generating accurate code to solve various problems. For instance, Liu et al. (2024b) uses LLMs as optimizers to directly generate new trial solutions via in-context learning. Similarly, FunSearch (Romera-Paredes et al., 2024) uses LLMs to guide evolutionary procedures in mathematical discovery. Furthermore, Veličković et al. (2024) extend FunSearch by evolving scoring functions, significantly enhancing performance on combinatorial competitive programming tasks. Liu et al. (2024a) propose to use LLM with evolutionary computation methods for automatic heuristic design. Recent work has used LLMs to generate heuristic functions in Rust or Python for symbolic planning, later employed by PDDL solvers (Tuisov et al., 2025; Corrêa et al., 2025). In contrast, we study heuristics within LLM-based inference-time search methods (Yao et al., 2024; Hao et al., 2023), aiming to improve decision making during search.

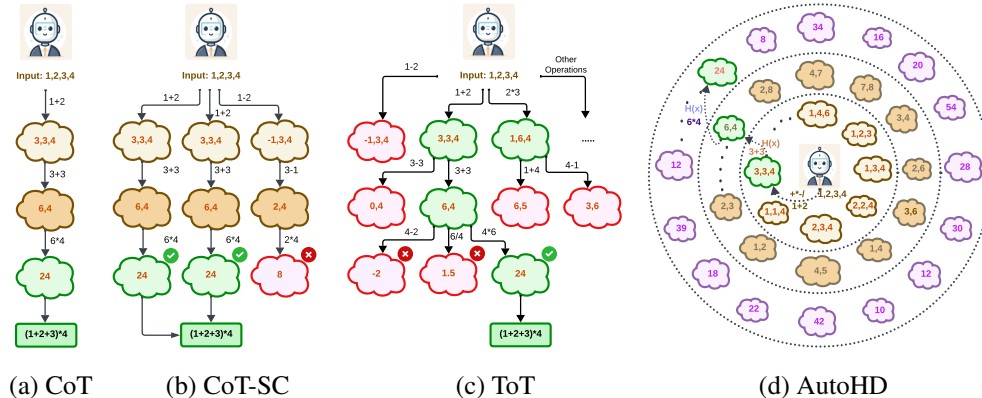

(a) CoT  (b) CoT-SC  (c) ToT  (d) AutoHD

Figure 2: Comparison between existing methods and the proposed method. (a) CoT follows a single linear path, thereby constraining its exploratory capacity. (b) CoT-SC extends this approach by performing multiple CoT iterations, leading to a result with higher confidence scores. (c) ToT introduces a tree-based search mechanism, branching systematically through intermediate states to explore a broader solution space. (D) In contrast, our AutoHD uses a heuristic function generated by the LLM to guide exploration. The heuristic prioritizes promising states, enabling more efficient and effective navigation of the solution space.

To this end, we use evolutionary algorithms to discover heuristics for symbolic problems, enhancing LLM performance on them.

## 3 AUTOMATED HEURISTICS DISCOVERY

Existing methods rely on LLMs to either self-verify or use additional models to evaluate the quality of each intermediate state $s_i$ (Yao et al., 2024; Ding et al., 2023). However, prior research (Huang et al., 2023; Stechly et al., 2024) has shown that self-verification is often unreliable. Meanwhile, additional model training requires significant costs in terms of data and computational resources. To overcome these challenges, we propose AutoHD, a novel approach that enables LLMs to explicitly generate heuristic functions to guide inference-

```python
def calc_heuristic(initial_state, final_state):
    initial_blocks = initial_state.split(', ')
    final_blocks = final_state.split(', ')

    misplaced_count = 0
    for initial, final in zip(initial_blocks, final_blocks):
        if initial != final:
            misplaced_count += 1

    heuristic_val = misplaced_count
    return heuristic_val
```

Figure 1: An example heuristic function proposed by LLMs for Blocksworld. The heuristic function computes the number of misplaced blocks and the cumulative positional differences of these blocks. The resulting sum provides an estimation of the discrepancy between states.

time search. Using the exceptional code generation capabilities of LLMs, AutoHD generates heuristic functions as Python code, enabling more accurate evaluation of intermediate states. To further improve the performance and robustness of these heuristic functions, we introduce a heuristic evolution process that iteratively refines them. Our framework reduces the inherent randomness of self-verification and enhances interpretability by providing explicit reasoning for why LLMs prioritize certain states. See Figure 2 for a comparison between our proposed method and existing approaches.

### 3.1 HEURISTIC FUNCTION PROPOSAL

A diverse set of initial heuristic functions is generated by prompting a pre-trained LLM, offering a flexible and automated solution that avoids reliance on hand-crafted components or additional models. The LLM generates both a natural language description and the corresponding Python code for each heuristic. The natural language description provides an intuitive, high-level overview of the heuristic function, while the Python code specifies its detailed implementation. These generated heuristic functions are designed to evaluate the quality or proximity of a given state $s$ relative to the goal states $\mathcal{G}$. We denote the generated heuristic function as $H \in \mathcal{H} : \mathcal{S} \times 2^{\mathcal{S}} \to \mathbb{R}$, where $2^{\mathcal{S}}$ indicates the power set of the state space $\mathcal{S}$.

An example is provided in Figure 1, which demonstrates a heuristic function for the Blocks World problem. The heuristic function estimates the cost of transitioning from an initial to a goal state in the Blocks World problem. It counts the number of blocks not in their correct positions, capturing structural misalignment. Besides, it computes the cumulative absolute positional difference of misplaced blocks, reflecting the effort required to reorder them. By summing these two, the heuristic provides a scalable and interpretable estimate of the discrepancy between states. See Appendix F.2 for the detailed prompt used in our experiments.

## 3.2 HEURISTIC GUIDED INFERENCE-TIME SEARCH

Given an LLM-proposed heuristic function $H$, it efficiently guides the search process. This heuristic function determines the order in which states are explored, ensuring that the search algorithm prioritizes promising paths while pruning unimportant ones. In this way, the heuristic plays a crucial role in balancing between exploring new states and exploiting known promising paths. Specifically, at a state $s$, the LLM generates the corresponding action space $\mathcal{A}_\theta(s)$, which contains all feasible actions from that state. For a given action $a \in \mathcal{A}_\theta(s)$, the LLM predicts the resulting next state $s'$ as $s' = T_\theta(s, a)$. The proposed heuristic function $H$ then computes a heuristic value $v$ for the predicted state $s'$, providing an estimation of desirability. Formally, the heuristic values for all possible next states are computed as

$$v_i = H(T_\theta(s, a_i)), \quad \text{for } a_i \in \mathcal{A}_\theta(s).$$

These values, collectively represented as $\mathcal{V} = \{v_1, v_2, \cdots\}$, guide the search algorithm by prioritizing the exploration of states with lower heuristic values. The heuristic values can be integrated into various search algorithms, such as $A^*$, beam search, or greedy search, to guide the exploration process. These values provide an estimation of the potential value of states, enabling the algorithms to prioritize or prune states dynamically and adapt to the specific requirements of the search strategy. In our work, we explore two search algorithms below.

**Greedy Breadth-First Search.** Greedy breadth-first search (Greedy BFS) is a heuristic-driven search algorithm that combines the exploration of breadth-first search with greedy prioritization. At each step, it evaluates all states in the frontier, which is the set of all states that have been generated but not yet explored or expanded, and selects the one with the smallest heuristic value. In this way, Greedy BFS effectively chooses the most promising states at each step. Let $Q$ represent the frontier set at a given step. For each state $s' \in Q$, the algorithm evaluates the heuristic value $H(s')$. It then selects the next state to expand as $s = \operatorname{argmin}_{s' \in Q} H(s')$, where $H(s')$ is the heuristic value of state $s'$. The algorithm terminates when a goal state is reached or the search budget is exhausted. See Algorithm 2 for details.

**$A^*$ Search.** $A^*$ search is a widely used algorithm that balances exploration and exploitation using a composite cost function. It evaluates states based on the cumulative cost to reach the state $G(s)$, and the estimated cost to reach the goal $H(s)$. In our case, each step has a uniform cost, meaning the cumulative cost $G(s)$ represents the number of steps from the start state $s_0$ to the state $s$. Formally, for each step, $A^*$ maintains a priority queue of frontier nodes, which is the set of all states that have been generated but not yet expanded, sorted by the summed cost $F(s) = G(s) + H(s)$. At each step, the algorithm selects the node $s$ with the smallest $F(s)$ value as $s = \operatorname{argmin}_{s' \in Q} F(s')$. The search continues until a goal state is expanded or the search budget is reached. See Algorithm 3 for details.

## 3.3 HEURISTIC EVOLUTIONS

To further enhance the performance and robustness of the proposed heuristic functions, we follow Liu et al. (2024a); Romera-Paredes et al. (2024) to include a heuristic evolution mechanism. In this approach, LLMs are prompted to generate initial heuristic functions, and subsequent generations of heuristic functions are derived iteratively. At each generation, new heuristic functions are evaluated, and only the top-performing ones are retained for the next round. Specifically, LLMs are prompted to generate an initial pool of $b$ heuristic functions, denoted as $H_1^0, H_2^0, \cdots, H_b^0$. The detailed prompts for each problem are provided in Appendix F.2. A small validation set consisting of approximately 10-15 problem instances, similar in size to the in-context learning examples, is used to evaluate the quality of each heuristic function. In generation $i$, given the heuristic functions $H_1^{i-1}, H_2^{i-1}, \cdots, H_b^{i-1}$ from the previous generation, we use two strategies for generating new heuristic functions, including exploration and modification. The exploration strategy focuses on generating new heuristic functions,

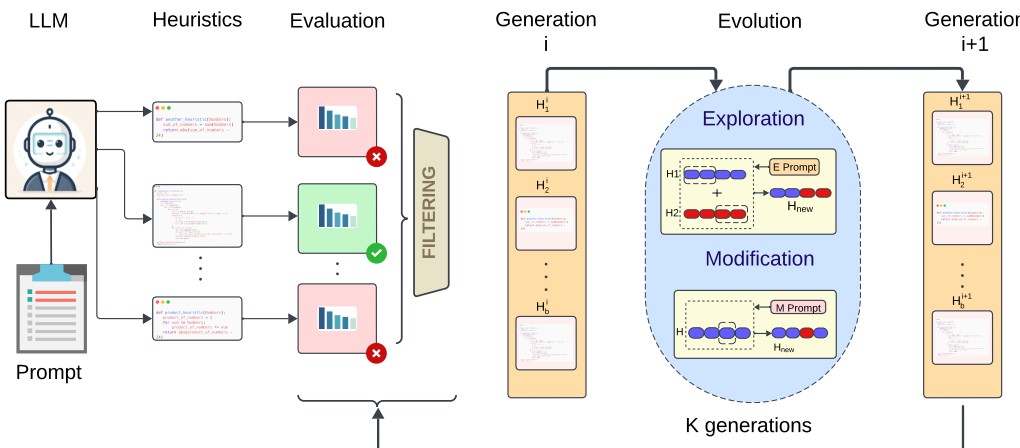

Figure 3: Heurisitc discovery process of the proposed method. The LLM is prompted to generate a diverse set of initial heuristic functions. These functions are evaluated on validation sets through heuristic-guided search to assess their quality. The top-performing heuristic functions are filtered and evolved to create the next generation by exploring new heuristic functions and refining existing ones. After K evolutions, the best heuristic function across all generations is selected for testing.

encouraging the exploration of diverse ideas. On the other hand, the modification strategy introduces minor variations to existing high-performing heuristic functions, such as changing parameters, to refine and improve them. This combination of strategies ensures both comprehensive exploration and efficient refinement of the heuristic function space. Details of the evolution prompts are provided in Appendix F.3. The newly generated heuristic functions $H_1^i, H_2^i, \cdots, H_b^i$ are then evaluated on the validation set through the heuristic-guided search process. The top-performing heuristic functions are selected to form the pool for the next generation. After several rounds of evolution, the best heuristic function across all generations is selected for testing. This process is performed only once, and no additional heuristic function generations are required during the inference-time search. The heuristic evolution process is summarized in Algorithm 1. See Figure 3 for an overview of the heuristic discovery process.

---

**Algorithm 1** Heuristic Evolution

---

Initialize a pool of $b$ heuristic functions $\mathbb{H}^0 = \{H_1^0, H_2^0, \cdots, H_b^0\}$
**for** $i \leftarrow 1$ to $K$ **do**
    Initialize an empty set $\mathbb{H}^i = \emptyset$
    $\mathbb{H}^i = \mathbb{H}^i \cup \text{Exploration}(\mathbb{H}^{i-1})$
    $\mathbb{H}^i = \mathbb{H}^i \cup \text{Modification}(\mathbb{H}^{i-1})$
    Evaluate each heuristic function $H \in \mathbb{H}^i$ on the validation set using heuristic-guided search
    $\mathbb{H}^i \leftarrow \text{sort}(\mathbb{H}^i, \text{validation accuracy})$
    $\mathbb{H}^i \leftarrow \mathbb{H}^i[: \lfloor |\mathbb{H}^i|/2 \rfloor]$
    $\mathbb{H}^i \leftarrow \text{sample}(\mathbb{H}^i, b)$
**end for**

---

## 3.4 DISCUSSIONS

Unlike existing methods that rely on LLM self-verification (Yao et al., 2024; Hao et al., 2023) or guidance from external models (Kambhampati et al., 2024), our approach uses LLMs to propose heuristic functions explicitly, which can guide the search process during inference. Additionally, while ToT (Yao et al., 2024) requires invoking LLMs to evaluate at every intermediate step during inference, our framework eliminates this overhead. Once the best heuristic function is identified on the validation set, no further heuristic function generation is necessary during inference. See Figure 2 for a comparison between our proposed method and existing approaches.

Previous works point out that an LLM can itself be treated as a heuristic. For instance, ToT (Yao et al., 2024) considers its own framework as a heuristic search algorithm, where the LLM serves as the heuristic by evaluating and ranking nodes during the search. However, prior methods (Stechly

et al., 2024; Huang et al., 2023) demonstrate that LLMs are inherently unable to self-verify, making them unreliable for providing robust heuristic values. Additionally, using the LLM itself to evaluate states lacks interpretability. This is because the LLM acts as a "black-box" evaluator, providing no insight into why certain states are preferred over others. As a result, the reasoning process becomes obscured, making it difficult to understand the rationale behind the LLM's decisions. In contrast, AutoHD enhances transparency by asking the LLM to generate these heuristic functions explicitly. This approach provides insight into the decision-making process, enabling a clearer understanding of why specific states are prioritized.

Additionally, some studies have explored process reward models (PRMs) (Luo et al., 2024; Li et al., 2022; Lightman et al., 2023), which enhance LLMs by evaluating and optimizing intermediate steps of reasoning or decision-making processes. It is worth noting that for symbolic problems, heuristic functions and PRMs serve a similar purpose, as both can guide the search. However, training PRMs often demands a large and diverse dataset of labeled intermediate states and their corresponding rewards, which can be challenging and costly to acquire, particularly for tasks with sparse data availability or where domain expertise is required to label the data (Lightman et al., 2023). In contrast, AutoHD shows that LLMs can generate heuristic functions on the fly, taking advantage of their pre-trained knowledge without requiring additional training or manual data collection. This efficiency makes LLM-proposed heuristic functions a desirable alternative to traditional PRMs in many applications.

## 4 EXPERIMENTS

In this section, we evaluate AutoHD on three tasks, including Blocksworld (Valmeekam et al., 2022), Game of 24 (Yao et al., 2024), and Rubik's Cube (Ding et al., 2023). Experimental results demonstrate that AutoHD significantly outperforms various baseline approaches. Additionally, we present an extensive ablation study in Section 4.4 to analyze the contributions of individual components. More experimental results can be found in Appendix A.

### 4.1 BLOCKSWORLD

Blocksworld (Valmeekam et al., 2022)involves a set of blocks, each with unique identifiers, which can be moved or stacked according to specific rules. The goal is to transform an initial configuration of blocks into a specified target configuration using a series of predefined actions. Each action moves blocks while adhering to spatial constraints. States represent the current arrangement of blocks, while transitions occur as actions are applied, updating the state to reflect changes in block positions. LLMs need to demonstrate spatial reasoning capabilities to understand the physical interactions and constraints inherent in the task.

We compare the proposed method with the following baseline methods, including (1) IO, which uses a standard input-output prompt; (2) CoT (Wei et al., 2022); (3) CoT-SC (Wang et al., 2022), an extension of CoT that selects answers via majority voting across multiple paths; (4) ToT (Yao et al., 2024), which use tree search to explore and expand intermediate steps; (5) RAP (Hao et al., 2023), which integrates Monte Carlo Tree Search (MCTS) with the LLM as a world model, rewarding intermediate steps and guiding tree growth toward the correct answer; (6) LLM-Modulo (Kambhampati et al., 2024), which uses external critics, verifiers, and human input to ensure correctness of generated plans; and (7) AoT (Sel et al., 2023), which provides sequences of intermediate steps as in-context examples. Note that, we use 5 iterations for self-consistency. For O1 mini, we evaluate the model on a subset of Blockworld by randomly sampling 20 instances per step. We evaluate all methods using accuracy as the metric, which measures whether the action trace generated by LLMs successfully solves the task, i.e., rearranges the blocks to achieve the specified goal configuration.

The results presented in Table 1 demonstrate that AutoHD consistently outperforms all baseline approaches across various LLMs, including GPT 4o-mini, GPT 4o, and Llama 3.1 70B, achieving accuracies of 42.4%, 71.5%, and 59.1%, respectively. Notably, AutoHD achieves approximately twice the accuracy of the second-best baseline on GPT 4o and GPT 4o-mini. Additionally, while some baselines exhibit significant performance variance across different LLMs, AutoHD maintains robust and consistent results, highlighting the generalization ability of our approach. It is also worth noting that while O1 demonstrates near-perfect accuracy as an oracle model, AutoHD outperforms

Table 1: Comparisons between AutoHD and baselines on Blocksworld. The best results are shown in bold.

| Methods | GPT 4o-mini | GPT 4o | Llama 3.1 70B | O1 mini |
|---|---|---|---|---|
| IO | 8.8 | 21.2 | 23.9 | 48.3 |
| CoT | 18.4 | 37.5 | 23.9 | - |
| CoT-SC@5 | 21.2 | 41.5 | 30.7 | - |
| ToT | 23.1 | 12.4 | 4.6 | - |
| RAP | - | - | 51.0 | - |
| LLM-Modulo | - | 48.0 | 13.0 | - |
| AoT | - | 43.0 | 17.0 | - |
| AutoHD | **42.4** | **75.1** | **59.1** | - |

Table 2: Comparisons between AutoHD and baselines on Game of 24. The best results are shown in bold.

| Methods | GPT 4o-mini | GPT 4o | LLaMA 3.1 70B | O1 mini |
|---|---|---|---|---|
| IO | 9 | 8 | 3 | 77 |
| CoT | 13 | 14 | 8 | - |
| CoT SC @ 5 | 15 | 14 | 8 | - |
| ToT | 42 | 62 | 59 | - |
| AutoHD | **54** | **70** | **69** | - |

Table 3: Comparisons between AutoHD and baselines on Rubik's Cube. The best results are shown in bold.

| Methods | GPT 4o-mini | GPT 4o | Llama 3.1 70B | O1 mini |
|---|---|---|---|---|
| IO | 0.0 | 0.0 | 0.0 | 0.6 |
| CoT | 0.0 | 0.0 | 0.6 | - |
| CoT-SC@5 | 0.0 | 0.6 | 0.6 | - |
| ToT | 0.6 | 9.8 | 13.1 | - |
| XoT | 67.2 | 79.8 | 78.1 | - |
| AutoHD | **82.5** | **83.1** | **84.7** | - |

Table 4: Ablation of search algorithms using 4o-mini on Blocksworld, Game of 24, and Rubik's Cube.

| | Blocksworld | | | Game of 24 | Rubik's cube |
|---|---|---|---|---|---|
| | 2 steps | 4 steps | 6 steps | | |
| BFS | 88.9 | 61.0 | 40.0 | 54.0 | 80.9 |
| A* | 95.7 | 73.3 | 69.2 | 50.0 | 82.5 |

O1 mini, a state-of-the-art large reasoning model (LRM) (Valmeekam et al., 2024), further validating its effectiveness.

### 4.2 GAME OF 24

The Game of 24 is a classic mathematical puzzle. It involves a set of four integers, typically drawn from a standard deck of playing cards, which can be combined using basic arithmetic operations, including addition, subtraction, multiplication, and division. The goal is to manipulate these numbers through a sequence of valid operations to produce the target value of 24. Each state in the game represents the current set of intermediate results, while transitions occur as operations are applied, reducing the number of operands and updating the state. Constraints include the correct application of operations and adherence to mathematical rules, such as division by non-zero numbers. The Game of 24 is an NP-Complete problem and requires LLMs to use strong arithmetic reasoning to solve.

We compare AutoHD with the following baseline methods, including IO, CoT (Wei et al., 2022), CoT-SC (Wang et al., 2022), ToT (Yao et al., 2024), and AoT (Sel et al., 2023). Note that, we use 5 iterations for self-consistency. We evaluate all methods using accuracy as the metric, which measures whether the action trace generated by LLMs successfully solves the task, i.e., forms the target number 24 by correctly applying a sequence of mathematical operations.

The results are shown in Table 2. The proposed AutoHD outperforms all baselines consistently on three different LLM models. AutoHD achieves the highest accuracies of 54%, 70%, and 69% for GPT 4o-mini, GPT 4o, and LLaMA 3.1 70B, respectively, significantly outperforming baselines. Simpler methods such as IO and CoT exhibit limited capability, achieving accuracies of at most 15%. This highlights the challenges inherent in solving the Game of 24, which involves intricate numerical reasoning and arithmetic operations. Furthermore, AutoHD achieves results comparable to O1 mini, further demonstrating the strength and innovation of the proposed method.

### 4.3 RUBIK'S CUBE

Rubik's cube (Ding et al., 2023) is a well-known puzzle-solving benchmark involving a cube with six faces, each subdivided into four smaller squares of distinct colors. The goal is to transform an initial scrambled configuration into a target state where each face of the cube is uniformly colored. This is

achieved through a sequence of predefined rotational actions applied to the cube's faces. Each action corresponds to rotating one of the cube's layers along a specified axis, which alters the arrangement of colored squares adhering to the cube's structural constraints. States represent the current color arrangement of the cube, while transitions occur as rotations are executed, updating the state to reflect the new configuration. Note that Rubik's cube is an NP-complete problem and the maximum number of steps required to optimally solve the cube is four in this dataset.

We compare our AutoHD with the following baselines, including IO, CoT (Wei et al., 2022), CoT-SC (Wang et al., 2022), ToT (Yao et al., 2024), and XoT (Ding et al., 2023). XoT uses reinforcement learning and MCTS to incorporate external domain knowledge by training an extra policy model on 1,000 training samples. We randomly select 15 examples from the training dataset in Ding et al. (2023) to form the validation set used in the heuristic evolution process. *We follow prior work Ding et al. (2023); Yu et al. (2024); Parashar et al. (2025) in using a fixed state transition function instead of relying on LLMs to update states, as these studies demonstrate that LLMs struggle with modeling state transitions in this benchmark.* We evaluate all methods using accuracy as the metric, which measures whether the action trace generated by LLMs successfully solves the task, i.e., transforms the cube into the goal state where each face is uniformly colored.

Results in Table 3 highlight the advance of AutoHD compared to baseline approaches on the Rubik's Cube task across various LLMs, including GPT 4o-mini, GPT 4o, Llama 3.1 70B. Specifically, AutoHD achieves the highest accuracies of 82.5%, 83.1%, and 84.7% with GPT 4o-mini, GPT 4o, and Llama 3.1 70B, respectively, outperforming the strongest baseline, XoT, by substantial margins of 15.3%, 3.3%, and 6.6%. Notably, simpler methods such as IO, CoT, and ToT struggle to solve the Rubik's Cube effectively, achieving nearly zero accuracy in most cases. This underscores the inherent difficulty of tasks like the Rubik's Cube, which require a deep understanding of complex spatial transformations. Even O1, a state-of-the-art large reasoning model, achieves only around 1% accuracy, further emphasizing the challenges posed by the Rubik's Cube and the inability of existing LLMs to effectively capture spatial relationships.

## 4.4 ABLATION STUDIES

In this subsection, we conduct extensive ablation studies to analyze the contributions of different components.

**Comparing Search Algorithms.** We begin by evaluating the performance of different search methods within our proposed framework. As discussed in Section 3.2, in this work we study two search algorithms, including $A^*$ and greedy BFS. Experiments are conducted using GPT-4o mini on three tasks, namely Blocksworld, the Game of 24, and the Rubik's Cube. For Blocksworld, the dataset is divided into subsets based on the minimum number of actions required to solve each test case. Specifically, we evaluate different search methods on a subset corresponding to 2-step problems. The results, presented in Table 4, demonstrate that both $A^*$ and greedy BFS perform effectively within our framework, achieving comparable outcomes across different tasks. These findings imply that LLM-generated heuristics are effective in guiding the search process during inference, highlighting the potential for future work to explore additional search algorithms.

*Ablation for action generation and state transition. We investigate how varying both the capacity of the LLM (GPT-4o Mini vs. GPT-4o) or using code for action generation and state transition affects AutoHD's performance. As shown in Table 5, using a stronger LLM improves performance. Notably, even the smaller model is effective at proposing heuristics. To further probe this, we evaluate a fully code-based setup for actions and transitions, which yields the best performance. These results suggest that for LLMs to operate independently in such domains, accurate action prediction and world modeling remain the primary bottlenecks.*

**Heuristic Evolutions.** We also conduct experiments to analyze the evolution of heuristic functions. For each generation, we select the heuristic functions with the highest validation accuracy and evaluate their performance on the test dataset using GPT 4o-mini. The experiments are conducted on the Rubik's Cube dataset. The results in Figure 4 show that both validation and test accuracies initially improve significantly during the early generations. Furthermore, performance eventually plateaus, indicating a saturation point in the evolution process. These suggest that the evolutionary process effectively explores the space of heuristic functions, ultimately generating a robust and well-performing heuristic function.

Table 5: Ablation on action generation ($\mathcal{A}_\theta$) and state transition ($T_\theta$). On Blocksworld and Game of 24, LLMs struggle in these components; *while stronger LLMs show improvement, code-based action generation, and state transitions work best. Notably, results show that LLM-generated heuristics are effective across LLM sizes.*

| Heuristic generator | $\mathcal{A}_\theta$ & $T_\theta$ | Blocksworld | | | Game of 24 |
|---|---|---|---|---|---|
| | | 2 steps | 4 steps | 6 steps | |
| 4o-mini | 4o-mini | 95.7 | 73.3 | 69.2 | 54.0 |
| 4o-mini | 4o | 100.0 | 98.8 | 84.9 | 59.0 |
| 4o-mini | code | 100.0 | 100.0 | 100.0 | 100.0 |
| 4o | 4o | 97.8 | 96.4 | 92.8 | 70.0 |
| 4o | code | 100.0 | 100.0 | 100.0 | 100.0 |

Figure 4: Ablation study of heuristic evolution on Game of 24 (Yao et al., 2024) dataset. The evolutionary process improves validation and test accuracies by helping explore robust heuristic functions.

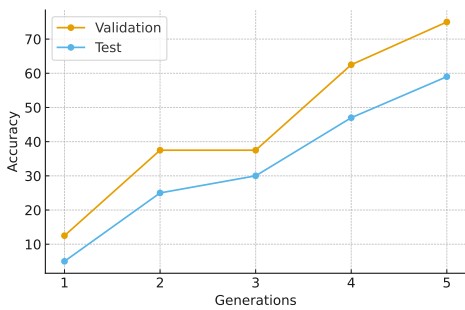

Table 6: Evaluation on Game of 24, with five candidate solutions per problem; correctness is counted if at least one is valid.

| | 4o-mini | 4o | LLaMA 70B |
|---|---|---|---|
| ToT | 47 | 68 | 61 |
| AutoHD | 66 | 92 | 86 |

Table 7: *Extension of AutoHD to Tree of Thought.*

| Method | Game of 24 |
|---|---|
| ToT | 42 |
| ToT + calculator | 46 |
| ToT + AutoHD | 52 |
| AutoHD | 54 |

**Multiple Solutions.** In symbolic problem-solving tasks, multiple solutions often exist for a single problem. To assess how AutoHD handles such cases, we evaluate it on the Game of 24. For each test instance, we generate five solutions and consider it correct if at least one is valid. As shown in Table 7, AutoHD finds multiple correct solutions. Unlike ToT, which relies on LLM self-evaluation during the search, the LLM-generated heuristic in AutoHD helps the LLM efficiently find multiple solutions.

*Orthogonal to Prior Work. AutoHD is complementary to existing inference-time techniques. To demonstrate this, we extend ToT on the Game of 24 task by enabling tool-use through a calculator (ToT + calculator) and an interpreter (ToT + interpreter). The interpreter setup allows us to integrate ToT with AutoHD by replacing the LLM-based state evaluation with the heuristic function generated by AutoHD. This simple substitution transforms ToT + interpreter into ToT + AutoHD, showing that AutoHD can be seamlessly integrated into existing baselines. Results in Table 7 illustrate this.*

## 5 CONCLUSIONS

In this paper, we propose AutoHD, a novel framework that enables LLMs to explicitly generate heuristic functions for guiding inference-time search. Moreover, the heuristic evolution process further refines these functions, enhancing their robustness and effectiveness. The proposed AutoHD requires no additional model training or fine-tuning, making it adaptable to various tasks. The explicit heuristic functions generated by LLMs provide valuable insights into the reasoning process, making AutoHD a transparent solution for complex decision-making. Extensive experimentation across diverse benchmarks has validated the efficacy of our approach, showing substantial performance improvements over multiple baselines. These results firmly establish AutoHD as a reliable and interpretable solution for addressing symbolic problem-solving tasks.

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

# Improving LLM Symbolic Problem-Solving via Automated Heuristic Discovery

# Appendix

## A  MORE EXPERIMENTAL RESULTS

In this section, we present additional experimental results. For the Blocksworld dataset, we divide the data into subsets based on the minimum number of actions required to solve each test case. The results are summarized in Table 8. *We also ablate heuristic functions generated by GPT-4o-mini within a symbolic planner, FastDownward (Helmert, 2006). Specifically, we use AutoHD to synthesize heuristic functions and then plug them into FastDownward for planning.*

Table 8: Comparisons between AutoHD and baselines on Blocksworld. The best results, with the exception of *Fast Downward + AutoHD* are shown in bold.

|  | Methods | Step 2 | Step 4 | Step 6 | Step 8 | Step 10 | Step 12 | All |
|---|---|---|---|---|---|---|---|---|
| GPT 4o-mini | IO | 46.7 | 25.0 | 5.9 | 0.7 | 0.0 | 0.0 | 8.8 |
|  | CoT | 38.3 | 26.7 | 26.5 | 13.1 | 8.0 | 0.0 | 18.4 |
|  | CoT-SC@5 | 55.3 | 36.1 | 30.3 | 12.4 | 4.4 | 0.0 | 21.2 |
|  | ToT | 71.1 | 39.3 | 26.3 | 16.6 | **20.5** | **17.4** | 23.1 |
|  | AutoHD | **95.7** | **73.3** | **69.2** | **20.9** | 8.0 | 0.0 | **42.4** |
| GPT 4o | IO | 51.1 | 39.3 | 30.3 | 13.9 | 0.9 | 2.2 | 21.2 |
|  | CoT | 68.1 | 45.4 | 52.3 | 28.1 | 19.5 | 19.6 | 37.5 |
|  | CoT-SC@5 | 63.8 | 39.5 | 57.4 | 35.3 | 26.5 | **28.2** | 41.5 |
|  | ToT | 75.5 | 21.4 | 10 | 1.3 | 2.7 | 2.2 | 12.4 |
|  | AutoHD | **97.8** | **96.4** | **92.8** | **80.8** | **42.9** | 15.2 | **75.1** |
| Llama 3.1 70B | IO | 44.4 | 32.1 | 30.9 | 19.9 | 13.4 | 4.4 | 23.9 |
|  | CoT | 46.7 | 44.1 | 28.3 | 19.9 | **18.8** | 4.4 | 26.1 |
|  | CoT-SC@5 | 42.2 | 47.0 | 39.5 | 22.5 | 17.4 | **19.6** | 30.7 |
|  | ToT | 60.0 | 0.0 | 0.0 | 0.0 | 0.0 | 0.0 | 4.6 |
|  | RAP | 67 | 76 | 74 | 48 | 17 | 9 | 51 |
|  | AutoHD | **95.6** | **96.4** | **82.9** | **55.6** | 13.4 | 0.0 | **59.1** |
| Symbolic Planner | Fast Downward + AutoHD | 100 | 100 | 100 | 100 | 100 | 100 | 100 |

### A.1  CHOICE OF HEURISTIC FUNCTIONS

In this subsection, we analyze the impact of heuristic function selection on performance. Specifically, we compare two strategies, including selecting the best heuristic functions across all generations and selecting the best heuristic functions from the final generation. Experiments are conducted using GPT 4o-mini. The results, presented in Table 9, indicate that LLMs explore various heuristic functions during the evolution process. Selecting the best heuristic function across all generations leads to more robust and stable performance compared to relying solely on the final generation.

Table 9: Ablation study on the choice of heuristic functions using GPT 4o-mini for Blocksworld, Game of 24, and Rubik's Cube.

|  | Blocksworld Step 2 | Game of 24 | Rubik's cube |
|---|---|---|---|
| All generations | 95.7 | 54 | 82.5 |
| Final generation | 93.3 | 41 | 65.6 |

Table 10: The number of states visited on the Blocksworld task of AutoHD and baselines.

| Method | Step 2 | Step 4 | Step 6 | Step 8 | Avg Cost |
|---|---|---|---|---|---|
| CoT | 4.5 | **5.3** | **6.7** | **9.1** | **6.4** |
| CoT SC@5 | 21.5 | 33.5 | 38.2 | 44.5 | 34.4 |
| ToT | 40 | 80 | 120 | 160 | 100 |
| RAP | 80 | 160 | 240 | 320 | 200 |
| LLM-Modulo | 20 | 40 | 60 | 80 | 50 |
| AoT | 4 | 8 | 12 | 16 | 10 |
| AutoHD | **3.2** | 14.1 | 43.2 | 97.5 | 39.6 |

### A.2 COST STUDY FOR AUTOHD

In this subsection, we study the average inference-time cost of AutoHD on the Blocksworld task compared to baselines. Specifically, we compute the average number of states visited using LLaMA-3 70B across all methods. The results in Table. 10 shows that AutoHD reduces the number of states visited during problem-solving. This efficiency stems from the use of the heuristic function to guide the search, allowing the model to focus on promising directions and terminate when appropriate. *As a result, Table 8 shows that AutoHD scales more robustly with increasing problem size compared to baselines.*

### A.3 OPTIMAL SOLUTION DISCOVERY VIA AUTOHD

We also study optimal solution results comparing baselines, using LLaMA 3.1 70B in this subsection. It is important to note that ToT and RAP construct a search tree with a **hardcoded maximum depth of $N$ steps**, where $N$ is set to the number of actions in the optimal solution, effectively injecting prior knowledge. In contrast, our approach introduces no such prior information. Instead, we rely solely on the heuristic function, terminating the search when the heuristic evaluates to 0. Remarkably, despite the absence of any injected knowledge about the optimal solution length, the results in Table 11 demonstrate that AutoHD consistently discovers optimal solutions, highlighting the effectiveness and precision of the heuristic in guiding the search.

*To further evaluate the quality of the generated heuristic functions, we test them for monotonicity and consistency, two standard properties in heuristic search. A heuristic is monotonous (or non-increasing) if its value does not increase along any path toward the goal. It is consistent if, for any state $s$ and successor $s'$, it satisfies $h(s) \leq c(s, s') + h(s')$, where $c(s, s')$ is the cost of transitioning from $s$ to $s'$ (Russell et al., 1995). These properties ensure that the heuristic never overestimates the remaining cost and guarantees steady progress during search.*

*We validate these properties by using the generated heuristic functions in two external planning setups: (i) a standard FastDownward (Helmert, 2006) symbolic planner for Blocksworld, and (ii) a code-based symbolic planner for Game of 24. In both cases, the planners consistently find the optimal plan using only the learned heuristic, indicating that it provides a smooth, goal-directed signal and satisfies monotonicity and consistency in practice.*

*Finally, in the few cases where the LLM fails to find an optimal solution, we observe that the failure typically stems from errors in action proposal or state transition modeling, rather than from limitations of the heuristic itself. This highlights the importance of accurate environment modeling for robust, end-to-end planning.*

### A.4 DATASETS

Rubik's Cube (Ding et al., 2023) is a well-known symbolic problem and puzzle-solving benchmark requiring multi-step spatial reasoning. The cube consists of six faces, each divided into four smaller squares, with each square assigned one of six distinct colors. The objective is to manipulate the cube from an initial scrambled state to a solved state, where each face is uniformly colored. This is accomplished through a sequence of predefined rotational moves applied to individual layers of the cube. Figure 5 provides a visual representation of the Rubik's Cube dataset used in this work. The

Table 11: Optimal solution discovery on Game of 24 and Blocksworld, comparing AutoHD with baselines. *We also integrate the heuristic functions found in AutoHD with FastDownward.*

| Method | Game of 24 | BW (2-step) | BW (4-step) | BW (6-step) | BW (8-step) |
|---|---|---|---|---|---|
| CoT | 4 | 4.5 | 5.3 | 6.7 | 9.1 |
| ToT | 4 | 2 | 4 | 6 | 8 |
| RAP | 4 | 2 | 4 | 6 | 8 |
| AutoHD | 4 | 2 | 4.1 | 6.1 | 8.1 |
| *FastDownward + AutoHD* | 4 | 2 | 4 | 6 | 8 |

left subfigure shows a scrambled cube configuration, representing an initial state in the dataset. The arrangement of colored squares encodes the current state of the puzzle, which requires a sequence of moves to reach the solved state. The right subfigure shows a goal state, where each face of the cube is uniformly colored. The dataset consists of cube states that are at most four moves away from the solved configuration.

## A.5 VALIDATION SET CONSTRUCTION

*To construct the validation set, we use the provided difficulty labels for tasks and randomly sample validation instances that span uniformly across all difficulty levels. To evaluate the impact of validation set size, we ablate different sizes and report the results over three random seeds on the Game of 24 dataset. This helps measure both average performance and variance during evolution.*

Table 12: *Effect of validation set size on accuracy and variability for Game of 24 (mean accuracy over three seeds).*

| Validation Set Size | Accuracy (%) | Std. Dev. |
|---|---|---|
| 5 | 40.00 | 7.21 |
| 10 | 53.00 | 4.58 |
| 15 | 56.00 | 3.00 |
| 20 | 57.33 | 2.08 |

*As expected, increasing the size of the validation set leads to lower variance and higher accuracy, as it offers better coverage for guiding the evolutionary process. In contrast, smaller validation sets result in higher variance and can lead to convergence toward suboptimal heuristic functions.*

## A.6 COST OF AUTOHD

*We compare the inference-time token usage and wall-clock cost of AutoHD and baseline methods across Game of 24, Blocksworld, and Cube.*

## B SEARCH ALGORITHM

In this section, we present the details of the search algorithms, specifically Greedy BFS and $A^*$, as outlined in Algorithm 2 and Algorithm 3, respectively.

## C HEURISTIC EVOLUTION DETAILS

We adopt an island-based evolution strategy inspired by Romera-Paredes et al. (2024). For each task, we first use the propose prompt to generate an initial pool of five candidate heuristic functions. These candidates are then evolved using the four different evolution prompts, producing 20 heuristic functions per iteration. Each function is evaluated on the validation set and ranked ($r$). We sample heuristic functions proportional to $r/N$, where $N = 20$ is the number of candidates in each iteration. This process is repeated for five iterations, after which the final heuristic is selected as the one with the best validation performance.

Table 13: *Inference-time token usage (in millions) and wall-clock time (hours) for evolution and search stages across tasks. The number of problems is indicated in brackets*

| Stage | Method | Game of 24 (100) | | Blocksworld (500) | | Cube (182) | |
|---|---|---|---|---|---|---|---|
| | | Tokens ($\times 10^6$) | Time (h) | Tokens ($\times 10^6$) | Time (h) | Tokens ($\times 10^6$) | Time (h) |
| Evolution | AutoHD | 0.52 | 1.2 | 3.30 | 14.0 | 0.74 | 1.3 |
| Search | CoT | 0.001 | 0.1 | 0.032 | 1.0 | 0.002 | 0.25 |
| | ToT | 2.000 | 8.0 | 1.300 | 9.0 | 0.220 | 2.0 |
| | RAP | 1.800 | 8.0 | 4.900 | 18.0 | – | – |
| | XoT | – | – | – | – | 0.420 | 3.2 |
| | AutoHD | 0.26 | 1.0 | 0.600 | 5.0 | 0.098 | 1.1 |

Table 14: An overview of datasets including Blocksworld, Game of 24, Rubik's Cube.

| | Blocksworld | Game of 24 | Rubik's Cube |
|---|---|---|---|
| Task | Propose actions to transform an initial block configuration to a goal | Use a list of four numbers to make 24, through +, -, $\times$ and / | Rotate a $2 \times 2$ Rubik's cube until each face has one single color |
| Input | A starting configuration of blocks | A starting list of 4 numbers | A scrambled $2 \times 2$ Rubik's cube |
| State | The current configuration of blocks | The updated list of numbers such as [1,2,3] | Updated colors on each face after a move |
| Action | Move, stack or unstack blocks | An arithmetic operation between any two numbers | One step rotation of a Rubik's cube face |
| Output | A sequence of actions | An equation evaluating to 24 for eg, $1 \times 2 \times 3 \times 4 = 24$ | The sequence of rotations to solve the Rubik's cube |

# D  HEURISTIC FUNCTIONS

In this section, we show some examples of heuristic functions generated by the LLMs. Table 15 shows some three representative examples generated by GPT 4o-mini. In the Blocksworld, the heuristic function estimates the effort required to transform an initial block configuration into a target configuration. It operates by first identifying the number of misplaced blocks that are not in their correct positions in the goal state. Additionally, it accounts for the cumulative positional difference, which measures how far each misplaced block is from its correct position. The final heuristic value is computed as the sum of these two terms. For the Game of 24, the heuristic function evaluates the proximity of a given set of numbers to the target value of 24. Given an input list of numbers, the heuristic iterates through all possible permutations of the numbers and arithmetic operations. The heuristic value is defined as the smallest absolute difference between 24 and the computed results of these expressions. This approach ensures that the heuristic effectively captures the validity of forming an expression that reaches 24. For the $2 \times 2$ Rubik's Cube, the heuristic function estimates the number of moves required to reach a solved state by examining the uniformity of the cube's faces. Specifically, it counts the number of faces where all four squares are of the same color. Since a

---

**Algorithm 2** Greedy BFS

**Require:** Initial state $s_0$, heuristic function $H(\cdot)$, action space $\mathcal{A}_\theta(\cdot)$, transition function $T_\theta(\cdot)$
$Q \leftarrow \{s_0\}$
**while** Q is not empty **do**
    $s \leftarrow \text{argmax}_{s' \in Q} H(s')$
    $Q \leftarrow Q \setminus \{s\}$
    **for** $a \in \mathcal{A}_\theta(s)$ **do**
        $s' \leftarrow T_\theta(s, a)$
        Return if $s'$ is a goal state
        $Q \leftarrow Q \cup \{s'\}$
    **end for**
**end while**

---

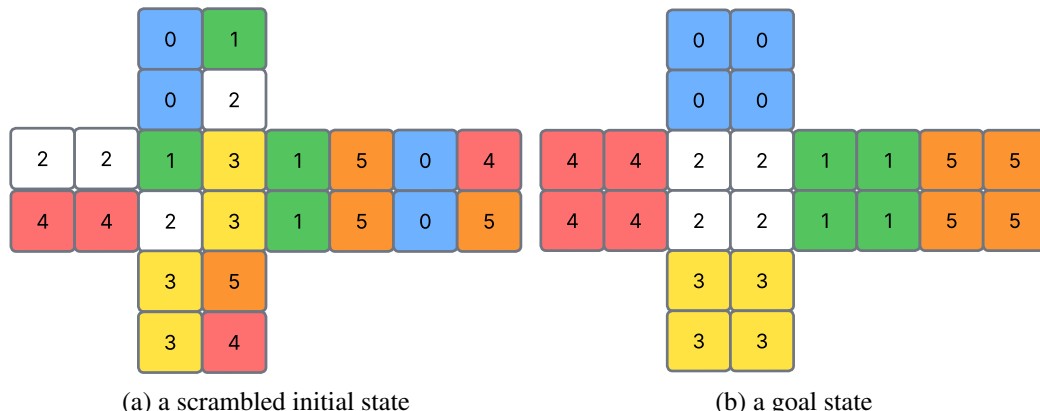

(a) a scrambled initial state       (b) a goal state

Figure 5: An example in Rubik's Cube dataset. The task is to transform a cube from a scrambled initial state to a goal state.

---

**Algorithm 3** $A^*$ Search

---

**Require:** Initial state $s_0$, heuristic function $H(\cdot)$, cost function $G(\cdot)$, action space $\mathcal{A}_\theta(\cdot)$, transition function $T_\theta(\cdot)$
$Q \leftarrow \{s_0\}$
**while** Q is not empty **do**
  $s \leftarrow \text{argmax}_{s' \in Q} H(s') + G(s')$
  $Q \leftarrow Q \setminus \{s\}$
  **for** $a \in \mathcal{A}_\theta(s)$ **do**
    $s' \leftarrow T_\theta(s, a)$
    Return if $s'$ is a goal state
    $Q \leftarrow Q \cup \{s'\}$
  **end for**
**end while**

---

completely solved cube has six uniform faces, the heuristic value is computed as the total number of faces, i.e. six, minus the count of uniform faces. This function serves as a coarse but effective estimate of the cube's disorder, guiding the search process toward configurations with increasing face uniformity. These examples illustrate the capacity of LLMs to generate domain-specific heuristic functions that align with problem-solving strategies typically designed by human experts.

# E  LLM USAGE

LLMs were used for text-refining purposes only.

# F  PROMPTS

In this section, we provide the detailed prompts used in our experiments.

## F.1  PROMPT TEMPLATE AND APPLICATION TO BENCHMARKS

*Our prompt template (Fig. 6) is inspired by prior work that integrates evolutionary algorithms with LLMs (Liu et al., 2024a; Romera-Paredes et al., 2024). It consists of three main components: the problem description, example problems, and previously generated heuristic functions. The problem descriptions and examples are taken directly from the dataset instructions of the benchmarks we evaluate on (Yao et al., 2024; Valmeekam et al., 2022; Ding et al., 2023). In iteration 0, there are no existing heuristic functions, so none are included in the prompt. Additionally, in this initial iteration, the evolution_type field is absent, and the task is framed as:* Please design a new heuristic. *In*

Table 15: Example heuristic functions generated by GPT 4o-mini.

| | Heuristic Functions | Explanations |
|---|---|---|
| Blocksworld | ```python
def calc_heuristic(initial_state, final_state):
    initial_blocks = initial_state.split(', ')
    final_blocks = final_state.split(', ')

    misplaced_count = 0
    for initial, final in zip(initial_blocks, final_blocks):
        if initial != final:
            misplaced_count += 1

    heuristic_val = misplaced_count
    return heuristic_val
``` | The heuristic function computes a heuristic value by summing the number of misplaced blocks and their cumulative positional differences. |
| Game of 24 | ```python
def calc_heuristic(Numbers):
    from itertools import permutations, product

    if len(Numbers) == 1:
        return abs(Numbers[0] - 24)

    operations = ['+', '-', '*', '/']
    heuristic_val = float('inf')

    for nums in permutations(Numbers):
        for ops in product(operations, repeat=len(nums)-1):
            expression = str(nums[0])
            for i in range(len(ops)):
                expression += f" {ops[i]} {nums[i+1]}"
            try:
                result = eval(expression)
                heuristic_val = min(heuristic_val, abs(result - 24))
            except ZeroDivisionError:
                continue

    return heuristic_val
``` | The function calculates the smallest absolute difference between 24 and the results of all possible arithmetic expressions formed using a given list of numbers and operations. |
| Rubik's Cube | ```python
def calc_heuristic(State):
    # Count the number of uniform faces
    uniform_face_count = 0

    for i in range(6):
        face = State[i * 4:(i + 1) * 4]
        if len(set(face)) == 1:  # Check if all squares on the face are the same color
            uniform_face_count += 1

    # Heuristic value is the total faces (6) minus uniform faces
    heuristic_val = 6 - uniform_face_count

    return heuristic_val
``` | This heuristic function estimates the cost to solve a $2 \times 2$ Rubik's Cube by counting the number of non-uniform faces on the cube. |

*subsequent iterations, we include heuristic functions generated in previous rounds, and the prompt specifies the evolution_type as either* `modification` *or* `exploration`, *prompting the model to either refine an existing heuristic or propose a novel one.*

## F.2    PROMPTS FOR HEURISTIC FUNCTION PROPOSAL

In this subsection, we show the final prompts (based on the template) we used to propose the initial heuristic functions For Blocksworld, Game of 24, and Rubik's Cube (Fig. 8, 9, and 10). *These prompts include the problem description and representative examples from the respective datasets, but omit prior heuristic functions, as in iteration 0, there are no heuristic functions yet.*

## F.3    PROMPTS FOR HEURISTIC EVOLUTION

*In this subsection, we show the final prompts (based on the template) we used for evolution of heuristics for Blocksworld, Game of 24, and Rubik's Cube (Fig. 11, 12, and 13).*

**Prompt Template for Heuristic Evolution**

I need help designing a new heuristic function to solve {Problem description}
{Problem examples} <exisiting_heuristics>

Task:
<evolution_type>
Firstly, identify the common idea in the provided heuristics.
Secondly, based on the backbone idea describe your new heuristic in one sentence.
Thirdly, implement it in Python as a function named 'calc_heuristic'. This function should accept 2 string inputs as shown above (comma separated with and at the last):
1. 'initial_state' - The current state.
2. 'final_state' - The final state.

This function should return one output: 'heuristic_val', which is the heuristic value calculated for the current state.

Do not give additional explanations.

Figure 6: Prompt Template for Heuristic Evolution

**Evolution Types**

1. Please help me create a new heuristic that has a totally different form from the given ones.
2. Please help me create a new heuristic that has a totally different form from the given ones but can be motivated from them.
3. Please assist me in creating a new heuristic that has a different form but can be a modified version of the heuristic provided.
4. Please identify the main heuristic parameters and assist me in creating a new heuristic that has a different parameter settings of the score function provided.

Figure 7: Evolution Types Prompts

1026
1027
1028
1029
1030
1031
1032
1033
1034
1035
1036
1037
1038
1039
1040
1041
1042
1043
1044
1045
1046
1047
1048
1049
1050
1051
1052
1053
1054
1055
1056
1057
1058
1059
1060
1061
1062
1063
1064
1065
1066
1067
1068
1069
1070
1071
1072
1073
1074
1075
1076
1077
1078
1079

**Prompt for Heuristic Function Proposal for Blocksworld**

I need help designing a new heuristic function to solve blocksworld problem. You are given 2 strings - current state of blocks and a final desired state. The goal is to design an appropriate heuristic that can be used to solve how far current state is from final state.

Here is an example problem:
Initial state: the red block is clear, the yellow block is clear, the hand is empty, the red block is on top of the blue block, the yellow block is on top of the orange block, the blue block is on the table and the orange block is on the table
Final state: the orange block is clear, the yellow block is clear, the hand is empty, the orange block is on top of the red block, the red block is on top of the blue block, the blue block is on the table, and the yellow block is on the table.

Here is another example problem:
Initial state: the blue block is clear, the orange block is in the hand, the red block is clear, the yellow block is clear, the hand is holding the orange block, the blue block is on the table, the red block is on the table, and the yellow block is on the table.
Final state: the orange block is clear, the red block is clear, the yellow block is clear, the hand is empty, the red block is on top of the blue block, the blue block is on the table, the orange block is on the table, and the yellow block is on the table.

Task:
Please design a new heuristic.
Firstly, describe your heuristic and main steps in one sentence. Start the sentence with 'Heuristic Description:'

Next, implement it in Python as a function named 'calc_heuristic'. This function should accept 2 string inputs as shown above (comma separated with and at the last):
1. 'initial_state' - The current state of blocks.
2. 'final_state' - The final state of blocks.

This function should return one output: 'heuristic_val', which is the heuristic value calculated for the current state of the blocks with respect to final goal state.

Do not give additional explanations.

Figure 8: Final Prompt for Heuristic Function Proposal for Blocksworld

**Prompt for Heuristic Function Proposal for Game of 24**

I need help designing a new heuristic function to solve Game 24 problem. In Game 24, you are given a list of numbers that need to be used with any operation '+', '-', '*', '/' to obtain the goal, which is [24]. You need to design an appropriate heuristic that can be used to solve how far current state is from final state.

Here is an example problem: Current State: [4, 4, 6, 8]  # (4 + 8) * (6 - 4) = 24
Final State: [24]

Here is another example problem: Current State: [4, 6] # 4 * 6 = 24
Final State: [24]

Here is another example problem: Current State: [8, 4, 1, 8]  # (8 / 4 + 1) * 8 = 24
Final State: [24]

Here is another example problem: Current State: [5, 5, 5, 9]  # 5 + 5 + 5 + 9 = 24
Final State: [24]

Here is another example problem: Current State: [24]
Final State: [24]

Task:
Please design a new heuristic.
Firstly, describe your heuristic and main steps in one sentence as a python comment. Start the comment with 'Heuristic Description:'

Next, implement it in Python as a function named 'calc_heuristic'. This function should accept 1 argument as show below, and not modify the input:
1. 'Numbers' - The current state, a list of numbers that has to be used to obtain the goal.

This function should return a single output, heuristic_val, representing the feasibility of reaching the goal, considering any operation using (+, -, *, /) with the current state. For eg: (a+b, a-b, b-a, a*b, a/b, b/a).
It must return 0 if the goal is achieved, i.e. when 24 is the only number left.

Do not give additional explanations.

Figure 9: Final Prompt for Heuristic Function Proposal for Game of 24

1134
1135
1136
1137
1138
1139
1140
1141
1142
1143
1144
1145
1146
1147
1148
1149
1150
1151
1152
1153
1154
1155
1156
1157
1158
1159
1160
1161
1162
1163
1164
1165
1166
1167
1168
1169
1170
1171
1172
1173
1174
1175
1176
1177
1178
1179
1180
1181
1182
1183
1184
1185
1186
1187

---

**Prompt for Heuristic Function Proposal for Rubik's Cube**

I need help designing a new heuristic function to solve 2x2 Pocket Cube. The problem is defined as the following. Your task is to restore a scrambled 2x2 Rubik's Cube to its original state. All the given problems can be solved in 1 to 4 moves. You cannot exceed more than 11 moves. Provide the sequence of moves required for the restoration. Please follow the instructions and rules below to complete the solving:

1. A 2x2 Pocket Cube has six faces, namely: [Upper, Front, Bottom, Left, Right, Back] Each consisting of a 2x2 grid of squares, with each square having its own color.

2. Colors in the Cube are represented in numbers: [0, 1, 2, 3, 4, 5]

3. The Cube's state is represented as an array of 24 elements. For instance, [0,0,0,0,1,1,1,1,2,2,2,2,3,3,3,3,4,4,4,4,5,5,5,5]. The Cube's state is represented as a 24-element array, where each group of 4 consecutive elements corresponds to a face of the cube in the following order: Upper face: Elements at indices 0 to 3. Right face: Elements at indices 4 to 7. Front face: Elements at indices 8 to 11. Down face: Elements at indices 12 to 15. Left face: Elements at indices 16 to 19. Back face: Elements at indices 20 to 23. Each element within a group represents the color or state of a specific square on that face.

4. A restoration of a Pocket Cube is to move squares in each face to have same numbers. Some example Restored States are [0,0,0,0,1,1,1,1,2,2,2,2,3,3,3,3,4,4,4,4,5,5,5,5].

You must make move to the Cube to achieve a Restored State, not limited to the above one. Note that we just need each face to have same numbers, no matter which face has which color.

Task:

Please design a new heuristic.

Firstly, describe your heuristic and main steps in one sentence. Start the sentence with 'Heuristic Description:'

Next, implement it in Python as a function named 'calc_heuristic'. This function should accept 1 input as shown above :
1. 'State' - The current state of 2x2 Cube, which is a numpy array.

This function should return one output: 'heuristic_val', which is the heuristic value calculated for the current state of the 2x2 Cube with respect to one of restored states.

Do not give additional explanations.

---

Figure 10: Final Prompt for Heuristic Function Proposal for Rubik's Cube

## Prompt for Heuristic Evolution for Blocksworld

I need help designing a new heuristic function to solve blocksworld problem. You are given 2 strings - current state of blocks and a final desired state. The goal is to design an appropriate heuristic that can be used to solve how far current state is from final state.

Here is an example problem:
Initial state: the red block is clear, the yellow block is clear, the hand is empty, the red block is on top of the blue block, the yellow block is on top of the orange block, the blue block is on the table and the orange block is on the table
Final state: the orange block is clear, the yellow block is clear, the hand is empty, the orange block is on top of the red block, the red block is on top of the blue block, the blue block is on the table, and the yellow block is on the table.

Here is another example problem:
Initial state: the blue block is clear, the orange block is in the hand, the red block is clear, the yellow block is clear, the hand is holding the orange block, the blue block is on the table, the red block is on the table, and the yellow block is on the table.
Final state: the orange block is clear, the red block is clear, the yellow block is clear, the hand is empty, the red block is on top of the blue block, the blue block is on the table, the orange block is on the table, and the yellow block is on the table.

<exisiting_heuristics>

Task:
<evolution_type>
Firstly, identify the common idea in the provided heuristics.
Secondly, based on the backbone idea describe your new heuristic in one sentence.
Thirdly, implement it in Python as a function named 'calc_heuristic'. This function should accept 2 string inputs as shown above (comma separated with and at the last):
1. 'initial_state' - The current state of blocks.
2. 'final_state' - The final state of blocks.

This function should return one output: 'heuristic_val', which is the heuristic value calculated for the current state of the blocks.

Do not give additional explanations.

Figure 11: Final Prompt for Heuristic Evolution for Blocksworld

1242
1243
1244
1245
1246
1247
1248
1249
1250
1251
1252
1253
1254
1255
1256
1257
1258
1259
1260
1261
1262
1263
1264
1265
1266
1267
1268
1269
1270
1271
1272
1273
1274
1275
1276
1277
1278
1279
1280
1281
1282
1283
1284
1285
1286
1287
1288
1289
1290
1291
1292
1293
1294
1295

**Prompt for Heuristic Evolution for Game of 24**

I need help designing a new heuristic function to solve Game 24 problem. In Game 24, you are given a list of numbers that need to be used with any operation '+', '-', '*', '/' to obtain the goal, which is [24]. You need to design an appropriate heuristic that can be used to solve how far current state is from final state.

Here is an example problem: Current State: [4, 4, 6, 8]  # (4 + 8) * (6 - 4) = 24
Final State: [24]

Here is another example problem: Current State: [4, 6]  # 4 * 6 = 24
Final State: [24]

Here is another example problem: Current State: [8, 4, 1, 8]  # (8 / 4 + 1) * 8 = 24
Final State: [24]

Here is another example problem: Current State: [5, 5, 5, 9]  # 5 + 5 + 5 + 9 = 24
Final State: [24]

Here is another example problem: Current State: [24]
Final State: [24]

<exisiting_heuristics>

Task:
<evolution_type>

Firstly, identify the common idea in the provided heuristics in one sentence as python comment. Start the python comment with 'Common Idea:'
Secondly, based on the backbone idea describe your new heuristic in one sentence as a python comment. Start the python comment with 'Heuristic Description:'
Thirdly, implement it in Python as a function named 'calc_heuristic'. This function should accept 1 argument as show below:
1. 'Numbers' - The current state, a list of numbers that has to be used to obtain the goal.

This function should return a single output, heuristic_val, representing the feasibility of reaching the goal, considering any operation using (+, -, *, /) with the current state. For eg: (a+b, a-b, b-a, a*b, a/b, b/a).
It must return 0 if the goal is achieved, i.e. when 24 is the only number left.

Do not give additional explanations. Do not use any tools. Return your response as python code.

Figure 12: Final Prompt for Heuristic Evolution for Game of 24

**Prompt for Heuristic Evolution for Rubik's Cube**

I need help designing a new heuristic function to solve 2x2 Pocket Cube. The problem is defined as the following. Your task is to restore a scrambled 2x2 Rubik's Cube to its original state. All the given problems can be solved in 1 to 4 moves. You cannot exceed more than 11 moves. Provide the sequence of moves required for the restoration. Please follow the instructions and rules below to complete the solving:

1. A 2x2 Pocket Cube has six faces, namely: [Upper, Front, Bottom, Left, Right, Back] Each consisting of a 2x2 grid of squares, with each square having its own color.

2. Colors in the Cube are represented in numbers: [0, 1, 2, 3, 4, 5]

3. The Cube's state is represented as an array of 24 elements. For instance, [0,0,0,0,1,1,1,1,2,2,2,2,3,3,3,3,4,4,4,4,5,5,5,5]. The Cube's state is represented as a 24-element array, where each group of 4 consecutive elements corresponds to a face of the cube in the following order: Upper face: Elements at indices 0 to 3. Right face: Elements at indices 4 to 7. Front face: Elements at indices 8 to 11. Down face: Elements at indices 12 to 15. Left face: Elements at indices 16 to 19. Back face: Elements at indices 20 to 23. Each element within a group represents the color or state of a specific square on that face.

4. A restoration of a Pocket Cube is to move squares in each face to have same numbers. Some example Restored States are [0,0,0,0,1,1,1,1,2,2,2,2,3,3,3,3,4,4,4,4,5,5,5,5].
You must make move to the Cube to achieve a Restored State, not limited to the above one. Note that we just need each face to have same numbers, no matter which face has which color.

<exisiting_heuristics>

Task:
<evolution_type>
Firstly, identify the common idea in the provided heuristics.
Secondly, based on the backbone idea describe your new heuristic in one sentence.
Thirdly, implement it in Python as a function named 'calc_heuristic'. This function should accept 1 input as shown above :
1. 'State' - The current state of 2x2 Cube, which is a numpy array.

This function should return one output: 'heuristic_val', which is the heuristic value calculated for the current state of the 2x2 Cube with respect to one of restored states.

Do not give additional explanations.

Figure 13: Final Prompt for Heuristic Evolution for Rubik's Cube

