# OpenReview forum: "Improving LLM Symbolic Problem-Solving via Automated Heuristic Discovery"
_ICLR.cc/2026/Conference — Submitted to ICLR 2026_

### Official Review · Reviewer_HKf4 · 2025-10-29

**Soundness:** 3
**Presentation:** 3
**Contribution:** 3
**Rating:** 8
**Confidence:** 4

**Summary:**

This paper introduces Automated Heuristic Discovery (AutoHD), a novel inference-time framework designed to enhance symbolic problem-solving using large language models (LLMs). While prior methods such as Chain-of-Thought (CoT) and Tree-of-Thought (ToT) enable intermediate-step exploration, they depend on unstable self-verification or costly external evaluators, often introducing noise and computational burden. AutoHD instead allows the LLM to generate its own heuristic functions—explicitly defined as executable Python code—to evaluate intermediate states during search, and iteratively refines these heuristics through an evolutionary process, improving robustness and accuracy. Without any fine-tuning or extra data, AutoHD produces interpretable, reliable reasoning across tasks like Blocksworld, Game of 24, and Rubik’s Cube, achieving up to 2× accuracy improvements over strong baselines.

**Strengths:**

Conceptual novelty: AutoHD moves beyond text-based reasoning chains by introducing explicit heuristic-driven search, letting LLMs quantify and guide their own symbolic reasoning.

Practical design: The framework operates entirely at inference time—no gradient updates, no retraining—making it lightweight and adaptable across tasks.

Interpretability: Each heuristic is represented as readable Python code, exposing the model’s decision logic and allowing transparent inspection of intermediate reasoning.

Evolutionary improvement: The iterative refinement of heuristics through evaluation and mutation leads to steadily improved accuracy and stability, reducing dependence on initial prompt quality.

Empirical coverage: Consistent superiority across multiple LLM scales and benchmarks—particularly in Rubik’s Cube (+6–15%) and Game of 24 (+25%) over ToT—validates its general effectiveness.

Strong preparation and reproducibility: Appendices detail all prompts, heuristic examples, and ablation setups, reflecting excellent completeness and methodological rigor.

Practical relevance: The work bridges symbolic AI and neural reasoning, providing a clear formalization of state-space search within LLM inference.

**Weaknesses:**

Ablation depth and integration:

- The paper lacks a consolidated component-wise ablation quantifying contributions of heuristic generation, heuristic-guided search, and heuristic evolution.
- Results are distributed across appendices rather than in the main text, making causal interpretation difficult.

Efficiency and computational overhead:

- The approach entails multiple LLM calls per inference and per evolution cycle; runtime and compute costs are not fully reported.
- There is no clear comparison of inference-time efficiency versus CoT/ToT under equal accuracy.

Scope and generalization:

- Only three symbolic tasks are tested. Extending to broader domains—e.g., Sokoban, text-based planning, or hybrid real-world tasks—would demonstrate stronger generality.
- Cross-task or transfer evaluations (e.g., reusing heuristics trained on one task for another) are not explored.

Comparative positioning:

- Although prior works on LLM-based automatic heuristic design (e.g., FunSearch, PRMs, LLM-PDDL) are discussed, empirical contrasts are missing.
- The paper would benefit from side-by-side comparisons highlighting differences in data requirements, compute cost, and interpretability.

Reproducibility and accessibility:

- Despite the detailed documentation, no public implementation or repository is provided. I recommend open-sourcing the code upon publication to maximize the paper’s transparency and impact.
- Sensitivity studies for key hyperparameters (generation number, pool size, rank threshold) and prompt stability are absent.

Statistical and failure analysis:

- Variance or confidence intervals are missing in main tables; failure case illustrations would clarify when heuristic evolution misguides or stagnates.

**Questions:**

Could the authors provide full component ablations (with/without heuristic, with/without evolution, different search algorithms) and integrate them into the main text?

How does the heuristic evolution quantitatively improve validation accuracy across generations beyond Fig. 4?

What is the runtime breakdown (number of LLM calls, total compute cost, wall-clock time) compared to CoT or ToT?

How does AutoHD scale with problem depth and branching factor in large symbolic tasks?

Can heuristics discovered in one domain (e.g., Blocksworld) transfer to related tasks or unseen instances?

How sensitive is AutoHD to prompt design and model scale (small vs. large LLMs)?

How does AutoHD differ empirically from LLM-generated planner heuristics (Tuisov et al., 2025; Correa et al., 2025) or Process Reward Models (PRMs) in terms of interpretability and inference efficiency?

Could an experiment combining AutoHD heuristics with a symbolic planner help isolate the contribution of inference-time integration?

---

> ### Author Response · Authors · 2025-11-21
> **Response to reviewer HKf4 1/2**
>
> We thank the reviewer for highlighting the many strengths of our paper; specifically, the reviewer points out the “**conceptual novelty**” of AutoHD in enabling heuristic-driven symbolic search, its “**practical design**” as an inference-only method, its “interpretability” via Python-code heuristics, and its “**empirical coverage**” with strong gains across tasks and models.
>
> Below, we address all the questions raised by the reviewer. We have also modified our manuscript according to these suggestions. Please note that all changes in the revised manuscript are highlighted in cyan and italicized for clarity.
>
> > Q1: On the requested ablations
>
> We respectfully point out that the requested ablations are already included in the paper. The first ablation, on the effect of heuristic functions, is captured by the comparison between AutoHD and the baselines, where AutoHD uses learned heuristic functions and the baselines rely on LLM self evaluation. The second ablation, on the effect of evolution, is shown in Figure 4, where iteration 1 corresponds to non evolved heuristics that perform poorly. Finally, we ablate different search algorithms in Tables 4 and 7, with Table 7 newly added in the revised manuscript.
>
> > Q2: On the performance gains from heuristic evolution
>
> In AutoHD, the heuristic function evolves over time through iterative refinement, with the best-performing functions retained for further mutation and evolution; this process steadily improves validation accuracy and yields more robust heuristics over successive generations
>
> > Q3: On the computational cost of AutoHD
>
> We provide the breakdown of the compute cost to run AutoHD vs baselines on the datasets. Inference-time token usage (in millions) and wall-clock time (hours) for evolution and search stages across tasks. The number of problems is indicated in brackets.
>
> | Stage     | Method | Game of 24 (100) Tokens (×10⁶) | Game of 24 (100) Time (h) | Blocksworld (500) Tokens (×10⁶) | Blocksworld (500) Time (h) | Cube (182) Tokens (×10⁶) | Cube (182) Time (h) |
> |-----------|--------|---------------------------------|----------------------------|----------------------------------|----------------------------|--------------------------|----------------------|
> | Evolution | AutoHD | 0.52                            | 1.2                        | 3.30                             | 14.0                       | 0.74                     | 1.3                  |
> | Search    | CoT    | 0.001                           | 0.1                        | 0.032                            | 1.0                        | 0.002                    | 0.25                 |
> |    | ToT    | 2.000                           | 8.0                        | 1.300                            | 9.0                        | 0.220                    | 2.0                  |
> |    | RAP    | 1.800                           | 8.0                        | 4.900                            | 18.0                       | --                       | --                   |
> |    | XoT    | --                              | --                         | --                               | --                         | 0.420                    | 3.2                  |
> |    | AutoHD | 0.26                            | 1.0                        | 0.600                            | 5.0                        | 0.098                    | 1.1                  |
>
>
> The cost of AutoHD’s evolution on a small validation set is a one-time cost, which helps save costs in search later, as heuristic functions guide the search more efficiently. We add this table to the Appendix.
>
> > Q4: On the application of AutoHD for large symbolic tasks
>
> We refer the reviewer to Tables 8 and 10 in our paper. Table 8 demonstrates that AutoHD remains robust on longer-horizon Blocksworld problems, outperforming baselines as the number of steps increases. Table 10 further supports this by showing that the evolved heuristic effectively guides the LLM toward promising states during search, significantly improving solution quality. As a result, AutoHD achieves up to 5× greater search efficiency compared to state-of-the-art baselines.
>
> We changed Section A.3 of the Appendix of our manuscript to make this clearer.

---

> ### Author Response · Authors · 2025-11-21
> **Response to reviewer HKf4 2/2**
>
> > Q5: On the generalization of AutoHD
>
> To demonstrate generalization, we introduce a new domain, Countdown [1], which can be viewed as a generalized version of the Game of 24, where the target is an arbitrary number (i.e., Game of X). We reuse the same heuristic function generated by AutoHD on Game of 24 and apply it directly to Countdown, observing comparable performance. This suggests that the learned heuristic transfers effectively to structurally similar tasks. We have included this result on GPT-4o-mini in the Appendix of our revised submission.
>
> | Method| Game of 24 | Countdown |
> |-----------|--------|---------------------------------|
> |ToT|                 42       |      36          |
> |AutoHD|           54       |      50         |
>
> > Q6: On the sensitivity of prompts in AutoHD across Language Models.
>
> We use the same prompts across all models and do not require any model-specific tuning (see Appendix F).
>
> > Q7: On the differences with (Tuisov et al., 2025; Correa et al., 2025) and PRMs
>
> Correa et al. (2025) and Tuisov et al. (2025) generate heuristics as explicit Python or Rust functions to explain symbolic planner decisions. In AutoHD, we instead use heuristic functions to interpret the black-box self-verification behavior of LLMs. Moreover, the inference optimization in Correa et al. (2025) and Tuisov et al. (2025) targets planner search cost, whereas AutoHD optimizes the inference-time search cost of LLMs. Finally, PRMs are trained black-box models and therefore do not offer the same level of interpretability.
>
> > Q8: On the combination of a symbolic planner with AutoHD
>
> Thanks for the valuable suggestion. We conduct experiments using heuristic functions produced by AutoHD within the symbolic planner FastDownward [2]. We add the FastDownward results in Tables 8 and 11, and observe that this setup solves all problems in the optimal number of steps. We revise the manuscript accordingly and update Appendix A.3
>
> References
>
> [1] Stream of Search (SoS): Learning to Search in Language, COLM, 2024
>
> [2] The Fast Downward Planning System, JAIR, 2006

---

> ### Author Response · Authors · 2025-11-27
>
> Dear Reviewer HKf4,
>
> As the discussion phase comes to a close, we sincerely thank you for your time and thoughtful feedback. We have carefully considered your comments and provided detailed responses to each of your concerns in our earlier reply.
>
> We greatly appreciate your assessment of our work, and please don’t hesitate to let us know if there are any remaining questions or points we can clarify.
>
> Warm regards,
>
> Authors

---

### Official Review · Reviewer_Y2Ed · 2025-10-30

**Soundness:** 4
**Presentation:** 3
**Contribution:** 3
**Rating:** 6
**Confidence:** 3

**Summary:**

The paper proposes AutoHD, which lets LLMs generate explicit heuristic fns as python code. This helps guide inference time search for symbolic problems. These functions are then iteratively refined in the paper via an evolutionary approach. The paper exhibits solid improvements over 3 tasks without model training.

The core idea is sound, but due to lack of theoretical results, generalizabiliity is questionable.

**Strengths:**

1. This approach is much more interpretable than black box LLM self verification methods.
2. The results are demonstrated across several tasks and LLMs, with nearly 2x improvements on some benchmarks.
3. The ablations are extensive, across equivalents like search algorithms, evolution heuristics, cost analysis, and comparison across LLMs.

**Weaknesses:**

1. There is no theoretical characterization of what makes a particular heuristic function good, and others bad.
2. The method seems computationally expensive since it needs multiple generations and multiple evals over validation set.
3. While the three symbolic domains chosen are interesting, it's a bit unclear if this generalizes to other symbolic domains as well.
4. The method seems too dependent on the distribution of the validation set, and does not adequately address how sensitive the approach is to quality or size of val sets. Also no empirical backing is provided for how to construct such sets in practice.

**Questions:**

1. How does the method perform when val set is scarce or not representative of the test data?
2. Are there biases or blind spots in the kinds of heuristics LLM generate?
3. What happens in methods where the search space is larger or the solution is a longer path?

---

> ### Author Response · Authors · 2025-11-21
> **Response to reviewer Y2Ed 1/2**
>
> We thank the reviewer for a positive evaluation of our work. Reviewer Y2Ed appreciates that our work **“exhibits solid improvements**” with “**nearly 2x improvement**” on some benchmarks. They also find our ablations as “**extensive**” and state as that the **“the core idea is solid”.**
>
> Below, we address all the questions and comments raised by the reviewer. We have also modified our manuscript according to these suggestions. Please note that all changes in the revised manuscript are highlighted in cyan and italicized for clarity.
>
> > Q1: On the Theoretical Characterization of a good heuristic function
>
> Great question! Theoretically, a heuristic function is considered effective if it satisfies the properties of monotonicity and consistency [1]. These properties ensure that the heuristic never overestimates the true cost to reach the goal from any given state. When these conditions are met, the search is assured to find an optimal solution in the minimal number of steps. We provide a more detailed discussion of these properties in our revised manuscript (Line 725 of Appendix A.3).
>
> To empirically validate our heuristic functions, we plug those generated by AutoHD (using GPT-4o-mini) into the symbolic planner FastDownward [2]. As shown in Table 11, the planner consistently finds optimal solutions across tasks, confirming that our heuristics are not only interpretable but also theoretically well-founded.
> | Method         | Game of 24 (4-step) | BW (2-step) | BW (4-step) | BW (6-step) | BW (8-step) |
> |----------------|------------|-------------|-------------|-------------|-------------|
> | FastDownward | 4          | 2           | 4           | 6           | 8           |
>
> > Q2: On the compute cost of AutoHD
>
> Below, we provide the breakdown of the compute cost to run AutoHD vs baselines on the datasets. Inference-time token usage (in millions) and wall-clock time (hours) for evolution and search stages across tasks. The number of problems is indicated in brackets.
>
> | Stage     | Method | Game of 24 (100) Tokens (×10⁶) | Game of 24 (100) Time (h) | Blocksworld (500) Tokens (×10⁶) | Blocksworld (500) Time (h) | Cube (182) Tokens (×10⁶) | Cube (182) Time (h) |
> |-----------|--------|---------------------------------|----------------------------|----------------------------------|----------------------------|--------------------------|----------------------|
> | Evolution | AutoHD | 0.52                            | 1.2                        | 3.30                             | 14.0                       | 0.74                     | 1.3                  |
> | Search    | CoT    | 0.001                           | 0.1                        | 0.032                            | 1.0                        | 0.002                    | 0.25                 |
> |    | ToT    | 2.000                           | 8.0                        | 1.300                            | 9.0                        | 0.220                    | 2.0                  |
> |    | RAP    | 1.800                           | 8.0                        | 4.900                            | 18.0                       | --                       | --                   |
> |    | XoT    | --                              | --                         | --                               | --                         | 0.420                    | 3.2                  |
> |    | AutoHD | 0.26                            | 1.0                        | 0.600                            | 5.0                        | 0.098                    | 1.1                  |
>
>
> The cost of AutoHD’s evolution on a small validation set is a one-time cost, which helps save costs in search later, as heuristic functions guide the search more efficiently. We add this table to the Appendix.
>
> > Q3: On the Generalization to other Symbolic Domains
>
> To demonstrate generalization, we use a new domain, Countdown [3], which can be viewed as a generalized version of the Game of 24, where the target is an arbitrary number (i.e., Game of X). We reuse the same heuristic function generated by AutoHD on Game of 24 and apply it directly to Countdown, observing comparable performance. This suggests that the learned heuristic transfers effectively to structurally similar tasks. We have included this result on GPT-4o-mini in the Appendix of our revised submission.
>
> | Method| Game of 24 | Countdown |
> |-----------|--------|---------------------------------|
> |ToT|                 42       |      36          |
> |AutoHD|           54       |      50         |
>
> For entirely different symbolic tasks, as long as the environment can be represented in terms of discrete states and transitions, we are confident that our method remains applicable.

---

> ### Author Response · Authors · 2025-11-21
> **Response to reviewer Y2Ed 2/2**
>
> > Q4: Concerns about the dependence of AutoHD on the validation set.
>
> When selecting the validation set, we utilize the provided difficulty labels for the tasks and randomly sample a validation set that spans uniformly across all difficulties. To test this mechanism, we ablate different sizes of the validation set across three different seeds, helping us get the standard deviation in performance for the Game of 24 dataset.
>
> | Size | Mean Acc Game of 24   | Std Dev |
> |------|--------|---------|
> | 5    | 40   | 7.21    |
> | 10   | 53.00  | 4.58    |
> | 15   | 56.00  | 3.00    |
> | 20   | 57.33  | 2.08    |
>
> As expected, with a larger validation set, the variance decreases and accuracy improves, as the dataset provides better coverage for guiding evolution. Conversely, with smaller validation sets, higher variance can lead to convergence toward suboptimal heuristic functions. We add this new experiment in Appendix Section A.5 in the revised manuscript.
>
> > Q5: Are there biases or blind spots in the kinds of heuristics LLMs generate?
>
> Thanks for the question! During the evolution process, a wide range of heuristic functions are generated, some of which have clear blind spots. For example, upon qualitative analysis of the Game of 24, some heuristics may initially over-prioritize operations like only multiplication or only addition. However, the evolutionary process naturally filters out these weaker heuristics over time, keeping only the ones that perform best for further evolution.
>
> > Q6: What happens when the search space is larger, or the solution is longer?
>
> We refer the reviewer to Tables 8 and 10 in our paper. Table 8 demonstrates that AutoHD remains robust on longer-horizon Blocksworld problems, outperforming baselines as the number of steps increases. Table 10 further supports this by showing that the evolved heuristic effectively guides the LLM toward promising states during search, significantly improving accuracy.
>
> We changed section A.3 of the Appendix of our manuscript to make this clearer.
>
> References
>
> [1] Artificial intelligence—a modern approach, Russel and Norwig, 1995
>
> [2] The Fast Downward Planning System, JAIR, 2006
>
> [3] Stream of Search (SoS): Learning to Search in Language, COLM, 2024

---

> ### Author Response · Authors · 2025-11-27
>
> Dear Reviewer Y2Ed,
>
> As the discussion phase comes to a close, we sincerely thank you for your time and thoughtful feedback. We have carefully considered your comments and provided detailed responses to each of your concerns in our earlier reply.
>
> We greatly appreciate your assessment of our work, and please don’t hesitate to let us know if there are any remaining questions or points we can clarify.
>
> Warm regards,
>
> Authors

---

### Official Review · Reviewer_1hNM · 2025-11-01

**Soundness:** 2
**Presentation:** 3
**Contribution:** 2
**Rating:** 4
**Confidence:** 4

**Summary:**

AutoHD is a training-free, test-time method that prompts an LLM to generate Python heuristic functions for symbolic problem solving, evolves those heuristics on a small validation set, and then guides search (greedy-BFS or A*) using the learned scores. The paper reports consistent gains on Blocksworld, Game of 24, and 2×2 Rubik’s, plus ablations on search choice, heuristic evolution, and swapping code vs LLM for action/transition modules.

**Strengths:**

1) Simple interpretable mechanism. Converting LLM suggestions into explicit code heuristics is simple, inspectable, and increases reliability.
2) Consistent empirical gains. The method improves over CoT/SC/ToT across tasks and beats XoT on Rubik’s in the reported setups
3) Search efficiency. Reporting states visited (not a complete cost analysis) indicates the approach prunes unpromising branches better than some baselines.

**Weaknesses:**

1) Baseline fairness. Baselines perform all computation using LLMs only, while AutoHD leverages external, code-based tools. A fair comparison requires search-based baselines with tool use enabled (e.g., ToT/ReAct + calculator/interpreter) under matched budgets.


2) Efficiency reporting incomplete. Results include states visited but omit end-to-end cost metrics (tokens, wall-clock, model calls) for both heuristic evolution and inference. Without these, practicality and cost–accuracy trade-offs against search-time scaling baselines remain unclear.

2) Domain dependence / prompt specificity. Appendix F prompts are highly tailored to each domain’s state encoding, indicating non-trivial overhead to adapt beyond the reported tasks. The method’s portability/generalizability is therefore limited without additional engineering.


3) Small validation for evolution. Heuristic selection on a very small validation set which can introduce selection bias, and the paper does not report run-to-run variance and stability analysis of the evolution procedure.


4) Cross-task application. The Rubik’s Cube evaluation uses a code-based transition or world model, while Blocksworld relies on LLM-generated transitions. This inconsistency in setup undermines methodological coherence and makes it unclear how the approach would generalize or be applied to a new domain.

**Questions:**

How were the 10 instances for heuristic evolution selected?

Do learned heuristics exhibit consistency/monotonicity along successful paths?

See weaknesses for additional analysis requested.

---

> ### Author Response · Authors · 2025-11-21
> **Response to Reviewer 1hNM 1/2**
>
> We thank the reviewer for their positive assessment of our work. They describe AutoHD as a “**training-free, test-time method**” that prompts an LLM to generate Python heuristics for symbolic problem solving. They highlight the **“simple, interpretable mechanism,**” noting that converting LLM suggestions into explicit code heuristics makes the method “**simple, inspectable, and increases reliability.**” The reviewer also emphasizes the “**consistent empirical gains**” and appreciates the “**search efficiency,**” observing that the method “**prunes unpromising branches better than some baselines**” based on the number of states visited.
>
> Below, we handle the limitations and questions raised by the reviewer. Please note that all changes in the revised submission are highlighted in cyan and italicized for clarity.
>
> > Q1: Evaluation of baselines with tool use for a fair comparison with AutoHD
>
> Great observation! To address this limitation, we conduct an experiment on the Game of 24 dataset using ToT, ToT + calculator, and ToT + interpreter. Notably, ToT + interpreter can be adapted into ToT + AutoHD by replacing the LLM-based evaluation at each state with our Python-based heuristic function generated through evolution. This demonstrates that AutoHD’s heuristic functions can be seamlessly integrated into existing search-based frameworks like ToT.
>
> | Method            | Game of 24 |
> |-------------------|------------|
> | ToT               | 42          |
> | ToT + calculator| 45     |
> | ToT + AutoHD  | 52      |
> | AutoHD            | 54      |
>
> We add this ablation in the main paper as Table 7 and explain the ablation in detail in L471.
>
> > Q2: On the compute cost of AutoHD
>
> Below, we provide the breakdown of the compute cost to run AutoHD vs baselines on the datasets for GPT-4o-mini. Inference-time token usage (in millions) and wall-clock time (hours) for evolution and search stages across tasks. The number of problems is indicated in brackets.
>
> | Stage     | Method | Game of 24 (100) Tokens (×10⁶) | Game of 24 (100) Time (h) | Blocksworld (500) Tokens (×10⁶) | Blocksworld (500) Time (h) | Cube (182) Tokens (×10⁶) | Cube (182) Time (h) |
> |-----------|--------|---------------------------------|----------------------------|----------------------------------|----------------------------|--------------------------|----------------------|
> | Evolution | AutoHD | 0.52                            | 1.2                        | 3.30                             | 14.0                       | 0.74                     | 1.3                  |
> | Search    | CoT    | 0.001                           | 0.1                        | 0.032                            | 1.0                        | 0.002                    | 0.25                 |
> |    | ToT    | 2.000                           | 8.0                        | 1.300                            | 9.0                        | 0.220                    | 2.0                  |
> |    | RAP    | 1.800                           | 8.0                        | 4.900                            | 18.0                       | --                       | --                   |
> |    | XoT    | --                              | --                         | --                               | --                         | 0.420                    | 3.2                  |
> |    | AutoHD | 0.26                            | 1.0                        | 0.600                            | 5.0                        | 0.098                    | 1.1                  |
>
>
> The cost of AutoHD’s evolution on a small validation set is a one time cost, which helps save costs in search later, as heuristic functions guide the search more efficiently. We add this table in the Appendix.
>
> > Q3: AutoHD relies on domain-specific prompts, making it harder to apply to new tasks without extra work.
>
> We respectfully decline that AutoHD requires us to do non-trivial prompt engineering. We highlight that the prompts follow a Template (Fig. 6), which is inspired by previous work, such as FunSearch [1], which uses LLM+evolution for algorithmic discovery. As shown in the template, we directly incorporate problem descriptions and in-context examples already available in the benchmark tasks we evaluate on [2,3,4]. This eliminates the need for additional manual effort. As a result, AutoHD can adapt to new tasks with minimal prompt tuning, thanks to our modular prompt template.
>
> We have clarified the prompt structure in our revised draft. Specifically, we added a new subsection (Appendix F.1) that outlines how our prompt template is used across tasks. Additionally, we revised Appendix F.2 and F.3 to explicitly reference the use of this template in both the heuristic proposal and evolution stages.

---

> ### Author Response · Authors · 2025-11-21
> **Response to Reviewer 1hNM 2/2**
>
> > Q4: On the concern of validation set bias
>
> Thanks for raising this concern. While choosing the validation set, we use the provided difficulty labels of the tasks and randomly sample a validation set spanning uniformly over all difficulties. To test this mechanism, we ablate different sizes of the validation set across three different seeds, helping us get the standard deviation in performance for the Game of 24 dataset.
>
> | Size | Mean Acc Game of 24   | Std Dev |
> |------|--------|---------|
> | 5    | 40   | 7.21    |
> | 10   | 53.00  | 4.58    |
> | 15   | 56.00  | 3.00    |
> | 20   | 57.33  | 2.08    |
>
> As expected, with a larger validation set, the variance decreases and accuracy improves, as the dataset provides better coverage for guiding evolution. Conversely, with smaller validation sets, higher variance can lead to convergence toward suboptimal heuristic functions.
>
> We add this new experiment in Appendix Section A.5 in the revised manuscript.
>
> > Q5: Concerns of cross-task evaluation due to differing transition mechanisms in Blocksworld and Rubik’s Cube.
>
> Our objective is to demonstrate AutoHD’s applicability across a diverse range of symbolic tasks, and thus we choose Blocksworld and Rubik’s Cube as representative domains. As highlighted in prior work [4,5], Rubik’s Cube poses significant challenges for LLMs, particularly in modeling state transitions, so we follow the setup of [4] and use a code-based world model for it. Furthermore, we conduct an ablation on Blocksworld using a code-based transition model (Table 5). The results suggest that while LLMs are effective at proposing heuristic functions, they often struggle with modeling environment dynamics and predicting valid actions.
>
> In response to your feedback, we have updated the caption of Table 5. We also refine and rewrite the accompanying discussion starting at Line 423 to clarify this ablation.
>
> > Q6: How were the instances for the validation set to do heuristic evolution chosen?
>
> As shown in our response to Q4, we randomly select a validation set. For each dataset, we use the provided task difficulty labels and sample the validation set uniformly across all difficulty levels.
>
> > Q7: Do learned heuristics exhibit consistency/monotonicity along successful paths?
>
> Thank you for raising this point. We clarify that the evolved heuristic function is both consistent and monotonous, as evidenced in Table 11, where LLMs generate optimal solutions for nearly all problems they successfully solve in Blocksworld and Game of 24. By definition, this implies that the heuristic does not overestimate the cost, which decreases reliably toward the goal. In the few remaining cases where the solution is not optimal, we observe that the failure typically arises from errors in action proposal or state transition modeling, not from the heuristic itself
>
> To further support this, we include new results in the revised Table 11 using FastDownward [6], a symbolic planner guided solely by our AutoHD heuristic. It consistently finds optimal solutions, reinforcing that the heuristic provides a reliable, goal-directed signal. Relevant discussion and definitions have been added in the Appendix of the updated draft.
>
> FastDownward results with an AutoHD-generated heuristic from GPT-4o-mini.
>
> | Method         | Game of 24 (4-step) | BW (2-step) | BW (4-step) | BW (6-step) | BW (8-step) |
> |----------------|------------|-------------|-------------|-------------|-------------|
> | FastDownward | 4          | 2           | 4           | 6           | 8           |
>
> References
>
> [1] Mathematical discoveries from program search with large language models, Nature, 2023
>
> [2] Tree of Thoughts: Deliberate Problem Solving with Large Language Models, Neurips, 2023
>
> [3] On the planning abilities of large language models-a critical investigation, Neurips, 2023
>
> [4] Everything of Thoughts: Defying the Law of Penrose Triangle for Thought Generation, ACL, 2023
>
> [5] Inference-Time Computations for LLM Reasoning and Planning: A Benchmark and Insights, CORR, 2025
>
> [6] The Fast Downward Planning System, JAIR, 2006

---

> ### Author Response · Authors · 2025-11-27
>
> Dear Reviewer 1hNM,
>
> As the discussion phase comes to a close, we sincerely thank you for your time and thoughtful feedback. We have carefully considered your comments and provided detailed responses to each of your concerns in our earlier reply.
>
> We greatly appreciate your assessment of our work, and please don’t hesitate to let us know if there are any remaining questions or points we can clarify.
>
> Warm regards,
>
> Authors

---

### Official Review · Reviewer_gjvy · 2025-11-06

**Soundness:** 3
**Presentation:** 3
**Contribution:** 2
**Rating:** 4
**Confidence:** 3

**Summary:**

This paper proposed AutoHD, a novel approach that let LLMs to generate heuristics automatically without test-time inference, self-verification, and external model training. AutoHD shows a consistent stronger performance across multiple tasks.

**Strengths:**

1. Method does not require model training or self-verification
2. Consistent better performance across multiple task
3. Clearly written

**Weaknesses:**

### Major
1. My impression for heuristic discovery is asking the model to analyze the current state and the target state, am i understanding it correctly? If this is the case, what's the difference between this method and some other others that decompose the problem into subtasks then aggregate the answer? For instance, when I have the task of "where did I park my car," some heuristics could be "next to a van" or "close to the entrance." Then i just continually evolve my heuristics. But what's the difference between this one and "sub problem 1: where is it next to" and "sub problem 2: is it close to the entrance?"
2. Can you also compare token usage?

### Minor
1. Line 39, logic is a little bit flawed. You are saying there are many methods to improve test time scaling, such as early approach CoT. Maybe change the narrative a little bit.
2. Line 47, "approaches that require additional model training" needs citation.
3. Line 50, it's better to define what "heuristics" mean here in the high level. Either an examples or formal definition are fine.

**Questions:**

1. Page 17, prompt template for heuristic, how to generate heuristic at the first place?

---

> ### Author Response · Authors · 2025-11-21
> **Response to reviewer gjvy**
>
> We thank the reviewer for highlighting several strengths of our work. They note that our method “**does not require model training or self-verification**,” which simplifies deployment and broadens applicability. They also observe “**consistent better performance across multiple tasks**,” underscoring the robustness of our approach. Finally, we appreciate the comment that the paper is “**clearly written.**”
>
> Below, we handle the limitations and questions raised by the reviewer. Please note that all changes in the revised submission are highlighted in cyan and italicized for clarity.
>
> > Q1: Clarification about the role of heuristic functions vs. subproblem decomposition
>
> Heuristic functions and subproblem decomposition are fundamentally different. A heuristic function estimates how far a given state is from the final goal. For example, in the Game of 24, a simple heuristic could compute the absolute difference between the current value and 24. This estimate is used only to guide the search toward the goal. Concrete examples of heuristic functions generated via AutoHD are shown in
> In contrast, subproblem decomposition changes the structure of the problem by introducing intermediate goals. Using the same example, this would mean creating subgoals such as reaching 12, then 18, and only then 24. This is a different paradigm: it augments the goal rather than estimating progress toward it.
> AutoHD uses heuristics in the first sense, i.e., estimating the distance to the goal, and not subgoal creation or decomposition. To further clarify this in our manuscript, we add an example in the Introduction (Line 51) and a clarification in Line 56.
>
> > Q2: On the compute cost of AutoHD
>
> Below, we provide the breakdown of the compute cost to run AutoHD vs baselines on the datasets for GPT-4o-mini. Inference-time token usage (in millions) and wall-clock time (hours) for evolution and search stages across tasks. The number of problems is indicated in brackets.
>
> | Stage     | Method | Game of 24 (100) Tokens (×10⁶) | Game of 24 (100) Time (h) | Blocksworld (500) Tokens (×10⁶) | Blocksworld (500) Time (h) | Cube (182) Tokens (×10⁶) | Cube (182) Time (h) |
> |-----------|--------|---------------------------------|----------------------------|----------------------------------|----------------------------|--------------------------|----------------------|
> | Evolution | AutoHD | 0.52                            | 1.2                        | 3.30                             | 14.0                       | 0.74                     | 1.3                  |
> | Search    | CoT    | 0.001                           | 0.1                        | 0.032                            | 1.0                        | 0.002                    | 0.25                 |
> |    | ToT    | 2.000                           | 8.0                        | 1.300                            | 9.0                        | 0.220                    | 2.0                  |
> |    | RAP    | 1.800                           | 8.0                        | 4.900                            | 18.0                       | --                       | --                   |
> |    | XoT    | --                              | --                         | --                               | --                         | 0.420                    | 3.2                  |
> |    | AutoHD | 0.26                            | 1.0                        | 0.600                            | 5.0                        | 0.098                    | 1.1                  |
>
>
> The cost of AutoHD’s evolution on a small validation set is a one-time cost, which helps save costs in search later, as heuristic functions guide the search more efficiently. We add this table to the Appendix.
>
> > Q3: Suggested revisions to the writing in the Introduction
>
> We have made changes in the introduction of our manuscript, which are outlined below:
>
>  -  Line 39 has been reworded to improve clarity and strengthen the logical flow.
>
>  -  A citation has been added at Line 47.
>
>  -  An illustrative example has been added after Line 50 to clarify the meaning of heuristics in this context.
>
> > Q4: How is the initial heuristic generated (Page 17 prompt template)?
>
> Thank you for the question. In iteration 0, the same prompt template is used, but the section for existing heuristics is left empty. The model is instructed with the phrase “Please design a new heuristic”, prompting it to generate the initial heuristic based solely on the problem description and the provided problem examples from the benchmark. This serves as the starting point for the evolutionary process. We have clarified the prompt structure in our revised draft. Specifically, we added a new subsection (Appendix F.1) that outlines how our prompt template is used across tasks. Additionally, we revised Appendix F.2 and F.3 to explicitly reference the use of this template in both the heuristic proposal and evolution stages.

---

> > ### Author Response · Authors · 2025-11-27
> >
> > Dear Reviewer gjvy,
> >
> > As the discussion phase comes to a close, we sincerely thank you for your time and thoughtful feedback. We have carefully considered your comments and provided detailed responses to each of your concerns in our earlier reply.
> >
> > We greatly appreciate your assessment of our work, and please don’t hesitate to let us know if there are any remaining questions or points we can clarify.
> >
> > Warm regards,
> >
> > Authors

---

### Comment · Area_Chair_Y2Aa · 2025-12-01

Dear Authors,

Due to the unprecedented decision of ICLR to prevent AC-Reviewer discussion, I have taken the unprecedented decision to engage in an AC-Author discussion for this paper. Note that this will be a poor substitute for a real author-reviewer discussion, as I will not have the time to put the same rigor into reading the paper and following up with your answers. However, I did read the current version of the paper and the full discussion here. I have a few questions remaining:

## Remaining Reviewer Concerns

Q1. **(Reproducibility and accessibility)**: No public implementation or repository is provided. Additionally, there are no prompts detailing the vanilla evaluation on the tasks (not heuristic-related). Can the authors commit to releasing prompts and code?

Q2. **(Statistical and failure analysis)**: Variance or confidence intervals will be nice to see in the main tables, as well. Further, it will be nice to see some qualitative observations on error modes, on top of the confidence intervals.

## Additional Questions

Q3. Can the authors provide further details about the baseline methods they use, for example, the ToT hyperparameters (i.e., breadth), and other relevant ones for RAP, XoT, etc.? This would help with reproducibility and can be used to judge the methods appropriately.

Q4. [1] presents significantly higher results on the Rubik's cube task (non-zero for even GPT-3.5 with IO), and higher values across the ToT and XoT experiments with weaker models (GPT-4 vs GPT-4o and even o1-mini). What is the difference in the evaluation setup that causes this discrepancy? It would be good to show that the results aren't influenced by factors, such as formatting, or output parsing.

### References

[1] Ding, Ruomeng, et al. "Everything of thoughts: Defying the law of penrose triangle for thought generation." Findings of the Association for Computational Linguistics: ACL 2024. 2024.

---

### Meta-Review · Area_Chair_Y2Aa · 2026-01-10

**Summary:**

All in all, I find the paper not ready for publication. There are a few major contributing factors to this, as outlined in **Reviewer Concerns**. First, the paper doesn't position itself well enough w.r.t. prior work. In particular, most of the ideas of the paper are taken from prior papers as acknowledged in the paper's text, and there is not enough explanation of where the technical novelty of the work lies. This is exacerbated by the poorly described experimental section that shows only a subset of experiments on old LLMs. Further, as currently designed, the prompts used for the LLMs seem a bit too problem-specific and difficult to adapt to new problems, putting into question the generalizability of the method. Finally, I think, given the importance of theoretical properties of the heuristics proposed by the LLM, there should be a greater effort in understanding how well those are followed by the LLM.

**Reviewer Concerns:**

I provided a quick list of outstanding concerns on my preliminary AC review on 1 Dec. Here, I will comment on outstanding concerns after the authors' response from 2 Dec. Note that those are a result of my more careful consideration of the paper and reviews under less time pressure. While it is undesirable to bring up concerns I couldn't during AC review on 1 Dec, this was necessitated by the limited time between being assigned as AC to the paper and the end of the discussion period.

- **Prior work comparison (Reviewer HKf4)**
I am concerned about how AutoHD compares technically (in terms of methodology) to RAP[1], FunSearch[2], and follow-up work derived from FunSearch[3,4]. It is unclear to me, also, why there is little empirical comparison to these methods. Only RAP is evaluated only on Blocksworld puzzles, only on Llama 3 70B, without any explanation for the missing results. I found that RAP requires logit access, which can explain the decision of the authors to only compare on the open model, but this is an explanation that the authors should have provided in the paper.
- **Experimental setup (AC)**
The authors provide a random subset of experiments without justifying their decisions. In particular, different prior works are compared on different problem settings, and o1-mini results are only provided for direct prompting. Further, outside of the direct prompting of o1-mini, there is no consideration of reasoning models in the paper. Given that it is ICLR2026 and a long list of open source and closed source reasoning models are available, which increase the capabilities of non-reasoning models drastically (Deepseek R1, GPT-OSS, Qwen3-Coder, Kimi K2, Gemini 3 Pro, Antorpic Opus 4.5, GPT 5.X, GPT Codex, to name a few), including code-specific ones, I find it unreasonable to conduct experiments only with GPT 4o.
- **Unclear what is the difference between Fast Downward + AutoHD and vanilla AutoHD (AC)**
I am confused why the authors get different results from Fast Downward with the AutoHD heuristics implemented and vanilla AutoHD. I thought both methods were a simple implementation of A* on top of the heuristics proposed and evolved by AutoHD.
- **Choice of validation set (Reviewer 1hNM, Reviewer Y2Ed)**
While the authors compare how the size of the evaluation dataset affects their method, they should also make an ablation w.r.t. the complexity of the dataset. In the setup of the experiment in Table 12, they already assume a large dataset where one can pick a representative sample. On the other hand, in a real scenario, a person might design a set by hand, which will necessarily have less complexity than the diversely picked one the authors propose.
- **Uninformative figures (AC)**
I find Figures 2 and 3 to be confusing, sometimes misleading,  not informative enough, and not closely enough following the actual method proposed. I would recommend that the authors take more time in designing them.
- **Generality of prompts (Reviewer 1hNM, Reviewer HKf4)**
I somewhat agree with the reviewer that even the latest prompt description in Appendix F, the prompts seem a bit "engineered". In particular, the game of 24 asks specifically for a heuristic of 0 when 24 is reached, which is nowhere to be seen in other games. Further, while one can deduce which parts of the prompts need to be changed for a new game, I think the authors should spend more space in the paper describing what each of the holes in the template should represent on a high level and how one can adapt this to a particular new game.
- **Monotonicity and Consistency of Heuristics (Reviewer 1hNM, Reviewer Y2Ed)**
While the author's response casts some light on this, I think a few questions remain. i) Is prompting the model to produce consistent and monotonic heuristics (with a possible description of what those properties entail) fixing the failures in monotonicity?  ii) Have the authors conducted manual verification of these properties? I believe that non-monotonic heuristics still have the opportunity to produce optimal solutions in large sets of testing examples, and therefore I am scared that the proposed empirical validation might be just masking the issue.

[1] Hao, Shibo, et al. "Reasoning with language model is planning with world model." Proceedings of the 2023 Conference on Empirical Methods in Natural Language Processing. 2023.
[2] Romera-Paredes, Bernardino, et al. "Mathematical discoveries from program search with large language models." Nature 625.7995 (2024): 468-475.
[3] Veličković, Petar, et al. "Amplifying human performance in combinatorial competitive programming." arXiv preprint arXiv:2411.19744 (2024).
[4] Liu, Fei, et al. "Evolution of heuristics: Towards efficient automatic algorithm design using large language model." arXiv preprint arXiv:2401.02051 (2024).

**Reviewer Scores:**

- **Reviewer gjvy**
I have chosen to exclude the review due to poor quality
- **Reviewer 1hNM**
The authors do not completely resolve all reviewer's issues, but I think they answer some important qs like evaluation dataset dependency, runtime comparison, and baselines with tool use. I think the reviewer should have raised their score to a 6.
- **Reviewer Y2Ed**
The authors address some of the reviewer's questions, including runtime, evaluation dataset dependency, and generalizability of heuristics. I think, though, the reviewer would have maintained their original score of 6.
- **Reviewer HKf4**
The original score of the reviewer is unjustifiably high. That said, the reviewer has raised some good points that were well addressed by the authors. Based on both the original strengths and weaknesses, as well as the provided answers, I think the judgment of the reviewer should be in the range of 6 to 8.

---

### Decision · Program_Chairs · 2026-01-26

Reject